# Fasciclin 2 engages EGFR in an auto-stimulatory loop to promote imaginal disc cell proliferation in *Drosophila*

Emma Velasquez[1], Jose A. Gomez-Sanchez[1,2], Emmanuelle Donier[1], Carmen Grijota-Martinez[1¤], Hugo Cabedo[1,2], Luis Garcia-Alonso[1]*

1 Instituto de Neurociencias CSIC-UMH, Universidad Miguel Hernandez, Sant Joan d'Alacant, Alicante, Spain, 2 Instituto de Investigación Sanitaria y Biomédica de Alicante (ISABIAL), Alicante, Spain

¤ Current address: Departamento de Biología Celular, Universidad Complutense de Madrid, and Instituto de Investigaciones Biomedicas "Alberto Sols" CSIC-UAM, Universidad Autonoma de Madrid, Spain
* lgalonso@umh.es

**Data Availability Statement:** All relevant data are within the manuscript and its Supporting Information files.

## Abstract

How cell to cell interactions control local tissue growth to attain a species-specific organ size is a central question in developmental biology. The *Drosophila* Neural Cell Adhesion Molecule, Fasciclin 2, is expressed during the development of neural and epithelial organs. Fasciclin 2 is a homophilic-interaction protein that shows moderate levels of expression in the proliferating epithelia and high levels in the differentiating non-proliferative cells of imaginal discs. Genetic interactions and mosaic analyses reveal a cell autonomous requirement of Fasciclin 2 to promote cell proliferation in imaginal discs. This function is mediated by the EGFR, and indirectly involves the JNK and Hippo signaling pathways. We further show that Fasciclin 2 physically interacts with EGFR and that, in turn, EGFR activity promotes the cell autonomous expression of Fasciclin 2 during imaginal disc growth. We propose that this auto-stimulatory loop between EGFR and Fasciclin 2 is at the core of a cell to cell interaction mechanism that controls the amount of intercalary growth in imaginal discs.

## Author summary

A key problem in developmental biology is how species-specific organ size is determined. Control of organ growth occurs at different levels of organization, from the systemic to the cell to cell interaction level. During nervous system development cell contact interactions regulate axon growth. Here, we show that one of the cell adhesion molecules involved in controlling axon growth, the *Drosophila* NCAM ortholog Fasciclin 2, also controls epithelial organ growth and size. Fasciclin 2 is expressed in highly dynamic but moderate levels during cell proliferation in imaginal discs (precursor epithelial organs of the adult epidermis), and at much higher level in pre-differentiating and differentiating cells in imaginal discs. During imaginal disc growth cell interactions mediated by Fasciclin 2 promote Epidermal Growth Factor Receptor function and cell proliferation. In turn, Epidermal Growth Factor Receptor activity promotes Fasciclin 2 expression, creating a cell autonomous auto-stimulatory loop that maintains cell proliferation. This function of

**Funding:** This work has been supported by grants from: Ministerio de Ciencia e Innovacion, BFU2016-76295-R (MCIN/AEI/10.13039/501100011033 and by "ERDF a way of making Europe") to LGA; Generalitat Valenciana, Conselleria d'Educació, Investigació, Cultura i Esport, Prometeo 2021-027 to LGA; Ministerio de Ciencia y Tecnologia, SAF2004-06593 to LGA; Ministerio de Ciencia y Tecnologia, FP2001-2181 to EV. The funders had no role in study design, data collection and analysis, decision to publish, or preparation of the manuscript.

**Competing interests:** The authors have declared that no competing interests exist.

Fasciclin 2 is reciprocal to its reported function in pre-differentiating and differentiating cells in imaginal discs, where it acts as an Epidermal Growth Factor Receptor repressor. Our study suggests that the amount of Fasciclin 2 may determine a threshold to grow or stop growing during epithelial organ development.

## Introduction

Morphogenesis involves the generation of organs with a species-specific size, pattern and shape. Cell proliferation is tightly controlled in rate and space during organ development to ensure a correct morphogenesis but, at the same time, its control is flexible enough to accommodate to local perturbation. Control of growth occurs at the systemic, organ and tissue organization levels by the action of hormones, morphogens and cell interactions [1–4]. Classic work demonstrated that local cell interactions are key in controlling intercalary cell proliferation to attain the final pattern and correct number of cells in vertebrate and invertebrate organs [5]. Cell to cell contact interaction mechanisms can provide the high degree of precision required to maintain species-specific patterns of intercalary growth during morphogenesis.

NCAM and L1-CAM are homophilic cell adhesion proteins of the immunoglobulin superfamily (IgCAMs) that couple highly specific cell recognition and adhesion with the control of Receptor Tyrosine Kinase (RTK) signaling [6–9]. NCAM- and L1-CAM-type proteins play key roles during normal development and cancer progression [9–11]. Interestingly, their normal function in signaling can be dissociated from their role in cell adhesion [12,13]. The synergistic coincident action of these IgCAMs and diffusible ligands on the RTKs may allow for a tight and precise spatial control of local growth, not achievable by the simple action of diffusible signals [14]. In *Drosophila* Fasciclin 2 (Fas2) is the ortholog of vertebrate NCAM, and it is expressed in neural and epithelial tissues [15–17]. Gain-of-function (GOF) conditions of Fas2 can promote EGFR activity during axon growth [6]. In contrast, *fas2* loss-of-function (LOF) mutations have been reported to cause derepression of the EGFR during retinal differentiation [18], and to interact with *warts* to produce over-proliferation in the follicular epithelium of the ovary [19].

*Drosophila* imaginal discs are epithelial organ precursors of the adult epidermis. Cell proliferation during imaginal disc growth is controlled by competitive cell interactions that help ensure the constancy of organ size and shape [20]. We have analyzed the role of Fas2 during imaginal disc growth using coupled-MARCM [21] and FLP-OUT genetic mosaics of Fas2 LOF conditions. Our results show that Fas2 expression is required for growth in a cell autonomous manner. Clones of cells devoid of Fas2 proliferate less than normal and behave as poor competitors during imaginal disc development. We show that Fas2 function during imaginal disc growth is directly mediated by the EGFR, as revealed by the reduced ppERK expression in Fas2-deficient cells and the genetic interactions between *fas2* LOF conditions and EGFR mutations. Moreover, Fas2-deficient clone rescue analysis shows that EGFR and its effectors Ras and Raf specifically act to mediate Fas2 function. Co-immunoprecipitation experiments show that Fas2 physically binds EGFR in cultured cells. Remarkably, EGFR activity in turn promotes the cell autonomous expression of Fas2, indicating the existence of a self-stimulatory feedback loop between Fas2 expression and EGFR function in imaginal discs. This positive feedback may correspond with the previously proposed self-stimulatory loop of EGFR activity in the wing imaginal disc [22]. In addition, we found that deficits of Fas2 indirectly cause compensatory increased levels of JNK and Yki activity.

Our results reveal a functionality of Fas2 during imaginal disc growth that sharply contrasts with its role as EGFR repressor at its peak of expression during retinal differentiation [18]. They suggest a scenario where the amount of Fas2 may determine whether to grow or stop growing, acting as an expression level growth-switch activator or repressor of EGFR respectively.

## Results

### Loss of Fas2 causes a cell autonomous deficit of growth in imaginal discs

All epithelial cells in imaginal discs express Fas2 in a dynamic pattern [16] (Fig 1A and 1B). The maximum expression corresponds to differentiating or pre-differentiating structures, like veins, proneural clusters, sensory organ precursor cells, and the Morphogenetic Furrow and photoreceptors in the eye imaginal disc. Undifferentiated proliferating epithelial cells also express Fas2 at lower levels (Fig 1A and 1B). Individuals lacking Fas2 are lethal but whole *fas2*⁻ null organs can develop and differentiate epidermis in gynandromorphs [16] (S1A Fig). We analyzed *fas2* LOF conditions generated by either hypomorphic allele combination or by restricted *RNAi* expression in the wing or the eye imaginal disc. The hypomorphic mutant combination and the specific expression of two different *fas2* RNAis in the wing imaginal disc caused a graded size reduction of the adult wing (Fig 1C, quantified in Fig 1G). Fas2 RNAi expression in the eye disc caused a severe size reduction of the imaginal disc and the adult head (Fig 1D). Organ size reduction was due to a lower number of cells, and not to a reduced cell size, as revealed by either the normal spacing of trichomes in the adult wing (which mark each single epidermal cell) (Fig 1C) or the cell profiles in imaginal discs and pupal wings (Fig 1E and 1F).

To study the cellular requirement of Fas2 in the null condition, we generated *fas2*⁻ null cell clones during the 1ˢᵗ and 2ⁿᵈ larval stages of larval development using the coupled-MARCM technique. The majority of Fas2-deficient clones were absent, and those surviving were reduced in size compared to their own control twins, or wild type clones, in wing and leg imaginal discs (Figs 2A, 2E, S1B and S1D), eye disc (Fig 2C, left picture), pupal wing (Fig 2D), and the adult structures derived from imaginal discs (S1C, S1E and S1F Fig). Fas2-deficient clones were more normal in the adult abdomen (S1C Fig) and the larval brain (Fig 2C, left panel). The size of pupal wings or imaginal discs bearing coupled-MARCM *fas2*⁻ clones was the same than WT coupled-MARCM controls (Fig 2D and 2F), indicating that the loss of growth produced by the *fas2*⁻ clones was compensated by the growth of the normal cells to attain a correct final organ size. In some cases, *fas2*⁺ twin control clones in imaginal discs or their derivatives were very large (S1B and S1E Fig). Analysis of the mitotic index in surviving coupled-MARCM *fas2*⁻ clones revealed a significant decrease compared to WT or their control twin clones (Fig 2G). Interestingly, the normal neighbor cells closest to *fas2*⁻ clones displayed a reduced mitotic index as well (Figs 2G and S1D), consistent with the loss of Fas2 homophilic interaction hindering cell proliferation. The growth deficit of *fas2*⁻ clones was rescued by the expression of either the GPI-anchored (Fas2^GPI) or the trans-membrane (Fas2^TRM) isoform of the protein (Fig 2B and 2C right picture; S1J Fig). Thus, *fas2*⁻ deficient cells can proliferate and differentiate epidermis in whole *fas2*⁻ deficient organs [16] or cell clones in the abdomen (S1C Fig), but they are hampered to do so when developing along with other cells expressing Fas2 in the same imaginal disc. This behavior of *fas2*⁻ cells in clones is symptomatic of cell competition (reviewed in [23]). It strongly suggests that slow proliferating Fas2-deficient cells can be out-competed by normal neighbor cells expressing Fas2. To confirm the presence of cell competition, we induced Fas2-deficient clones using the Minute method [24], which slows the proliferation rate of all the cells in the organ with the exception of those in the clone.

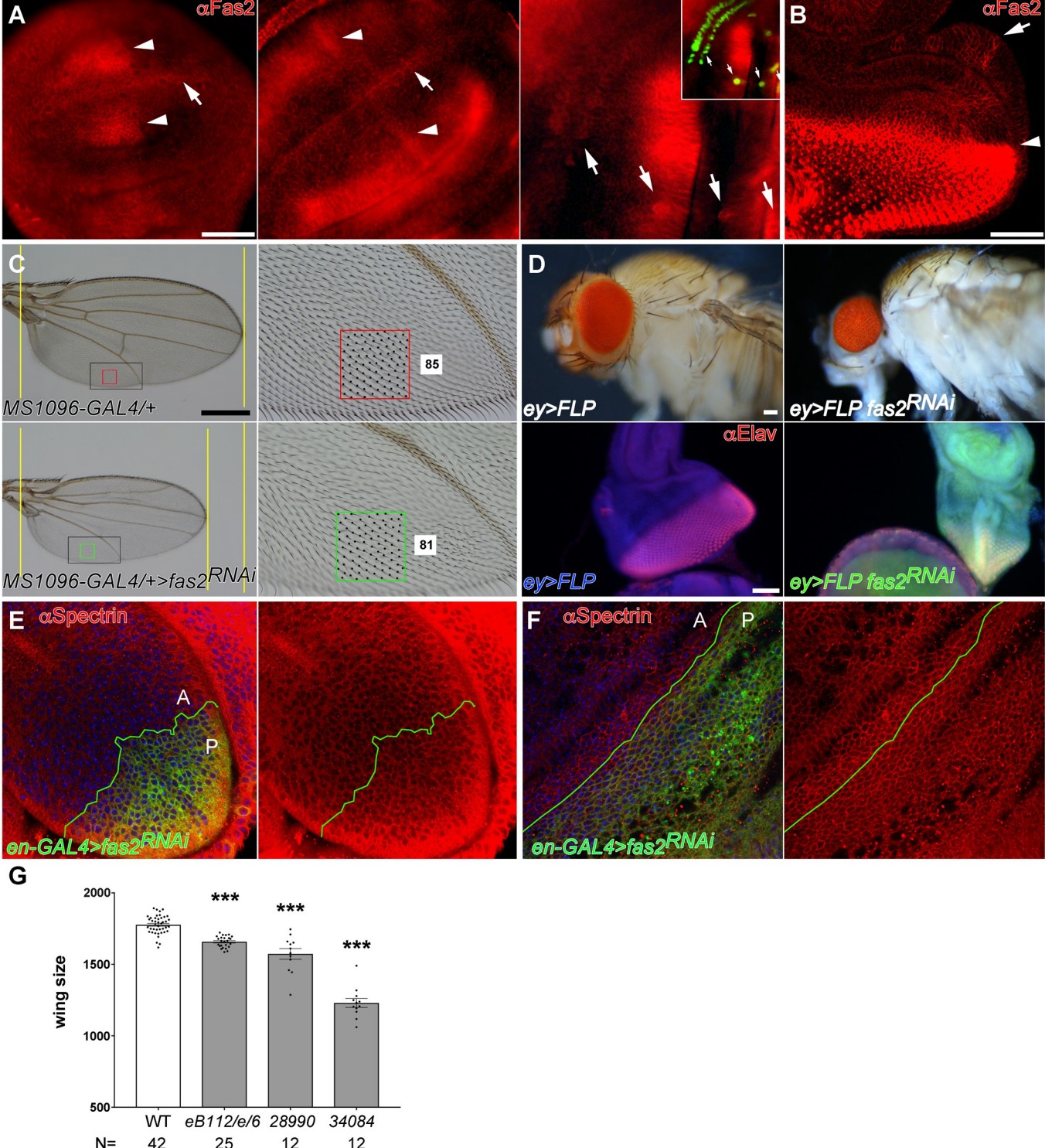

**Fig 1. Expression and requirement of Fas2 in imaginal discs.** (A) Fas2 expression (MAb-1D4 antibody) in early, middle and late stages of 3$^{rd}$ instar larva wing discs. Fas2 is expressed in all cells of the disc during all stages. Note the dynamic changing pattern of peaks of Fas2 expression at both sides (arrowheads) of the D/V compartment border (arrow) in early and middle 3$^{rd}$ instar wing discs (left and center). In late 3$^{rd}$ instar wing imaginal discs (at right) sensory organ precursor cells (arrows, marked with *neuralized-LacZ* in inset) express high levels of the protein. Bar: 50 µm. (B) Late 3$^{rd}$ instar larva eye-antenna imaginal disc. All cells express Fas2 in a dynamic pattern. The highest levels are found at the Morphogenetic Furrow (arrowhead), in the differentiating retina posterior to it and in the Ocellar prospective region (arrow). Bar: 50 µm. (C) Expression of *UAS-fas2$^{RNAi}$* (#34084) in the wing disc under the control of the *MS1096-GAL4* insertion

(doubly heterozygous females) causes a reduction of wing size (compare the two left panels), and loss of cross-veins. Size reduction is due to a lower number of epidermal cells in the *fas2^RNAi^* wing (compare the number of trichomes, each one representing a single cell, between the wing margin and the vein in the two right panels). Spacing and polarity of trichomes is normal in the *fas2^RNAi^* wing (85 vs. 81 cells inside the color squares). Bar: 500 μm. (D) Expression of *UAS-fas2^RNAi^* (#34084) in the eye-antenna imaginal disc in *eyeless*-driven FLP-OUTs (*ey>FLP*) causes a size reduction of the whole head (doubly heterozygous females, upper panels). The reduction in size is also evident in 3^rd^ instar larva eye-antenna imaginal discs (lower panels). At left, a control *ey-FLP; CyO/+; UAS-fas2^RNAi34084^*/+ sibling eye-antenna imaginal disc. At right, an *ey-FLP; ActinFRTy^+^FRT-GAL4 UAS-GFP/+; UAS-fas2^RNAi34084^*/+ eye-antenna imaginal disc. Note the severe reduction in size caused by the expression of *fas2^RNAi^*. Bar: 50 μm. (E) Expression of *UAS-fas2^RNAi^* (#34084) under the control of the *engrailed-GAL4* (*en-GAL4*) driver (*en-Gal4>fas2^RNAi^*) causes a reduction in the size of the posterior compartment of the wing imaginal disc. Shape and size of the cells is not affected by the expression of *fas2^RNAi^* (compare cell size in the anterior (A) and posterior (P) compartments). (F) Expression of *UAS-fas2^RNAi^* (#34084) in the P compartment of the wing under the control of the *en-GAL4* driver during pupal development does not alter cell size or shape. (G) Wing size in *fas2* LOF mutant combinations. WT controls are *MS1096-GAL4* heterozygous females. The hypomorphic *fas2^eB112^/fas2^e76^* combination causes a moderate but highly significant reduction of wing size, while the expression of *fas2^RNAi^*, RNAi #28990 or RNAi #34084, in *MS1096-GAL4* heterozygous females causes a stronger size reduction. Wing size is wing area in $\mu m^2/10^3$. N is number of individuals.

These *fas2^−^ Minute^+^* clones displayed an amelioration of the phenotype (Figs 2E, S1G, S1H and S1J), confirming that Fas2 is required for normal cell proliferation in imaginal discs.

## Fas2 insufficiency causes an indirect cell competition-dependent activation of the JNK pathway

Apoptosis of slow proliferating cells is a consequence of cell competition. *Minute^−^* heterozygous slow proliferating cell clones growing in a *Minute*-normal background can survive to the adult stage and differentiate epidermis only if induced during the 3^rd^ instar larval stage, but not when induced early in development [24]. In contrast, a significant number of early induced *fas2^−^* cell clones were able to evade apoptosis and differentiate in adult epidermis (S1F and S1J Fig). Apoptosis was found in a fraction of cells in *fas2^−^* cell clones (S1I Fig). To study the contribution of apoptosis to the *fas2* phenotype, we tested different genetic conditions that suppress cell death. Expression of the Drosophila-Inhibitor-of-Apoptosis (DIAP1, *UAS-diap*) [25] or induction of *fas2^−^* cell clones in a heterozygous *Df(3L)H99/+* background [26] produced a barely significant normalization of size in the *fas2^−^* clones, and we did not detect any significant effect at all expressing P35 in the clones (S1J Fig).

The JNK signaling pathway controls apoptosis [27] as well as cell proliferation and regeneration in imaginal discs [28–31], and interacts with Fas2 during neural differentiation in pupa [32]. To test the involvement of this signaling pathway in the *fas2* phenotype during imaginal disc growth, we induced the inhibition of JNK (Bsk) activity in the wing and eye disc of *fas2^RNAi^* LOF individuals. Wings and eyes respectively only displayed a partially corrected size, and they continued to be significantly smaller than their *bsk^RNAi^* controls (Figs 3A, 3B and S1K). Furthermore, coupled-MARCM *fas2^−^ bsk^RNAi^* clones continued being much smaller than their control twins (Fig 3C). Analysis of the expression of cleaved Caspase 3 in *en-GAL4 UAS-fas2^RNAi^* wing imaginal discs did not reveal obvious signs of apoptosis (Fig 3D).

To further analyze the contribution of the JNK pathway to the phenotype of Fas2-deficient cells, we monitorized the expression of the JNK pathway reporters *TRE-DsRed* [33] and *puckered-LacZ* (*puc-LacZ*) [34]. Expression of *TRE-DsRed* was increased in *en-GAL4 UAS-fas2^RNAi^* wing discs, indicative of JNK over-activation. Interestingly, this effect was non-cell autonomous and extended into the anterior compartment (Figs 3E and S2A, left), suggesting a compensatory/regenerative response in the organ [29,35]. Puc is a feedback repressor of the JNK pathway, and the *puc-LacZ* insertion causes a null mutation in the *puc* gene. Expression of *UAS-fas2 RNAi* in the posterior compartment of the wing disc using the *en-GAL4* driver caused a reduction of compartment size due to a reduced number of cells (Fig 1E), but no widespread signs of apoptosis (Fig 3D). The introduction of the heterozygous *puc-LacZ* insertion in this genotype produced a strong expression of the reporter (Figs 4A and S2A,

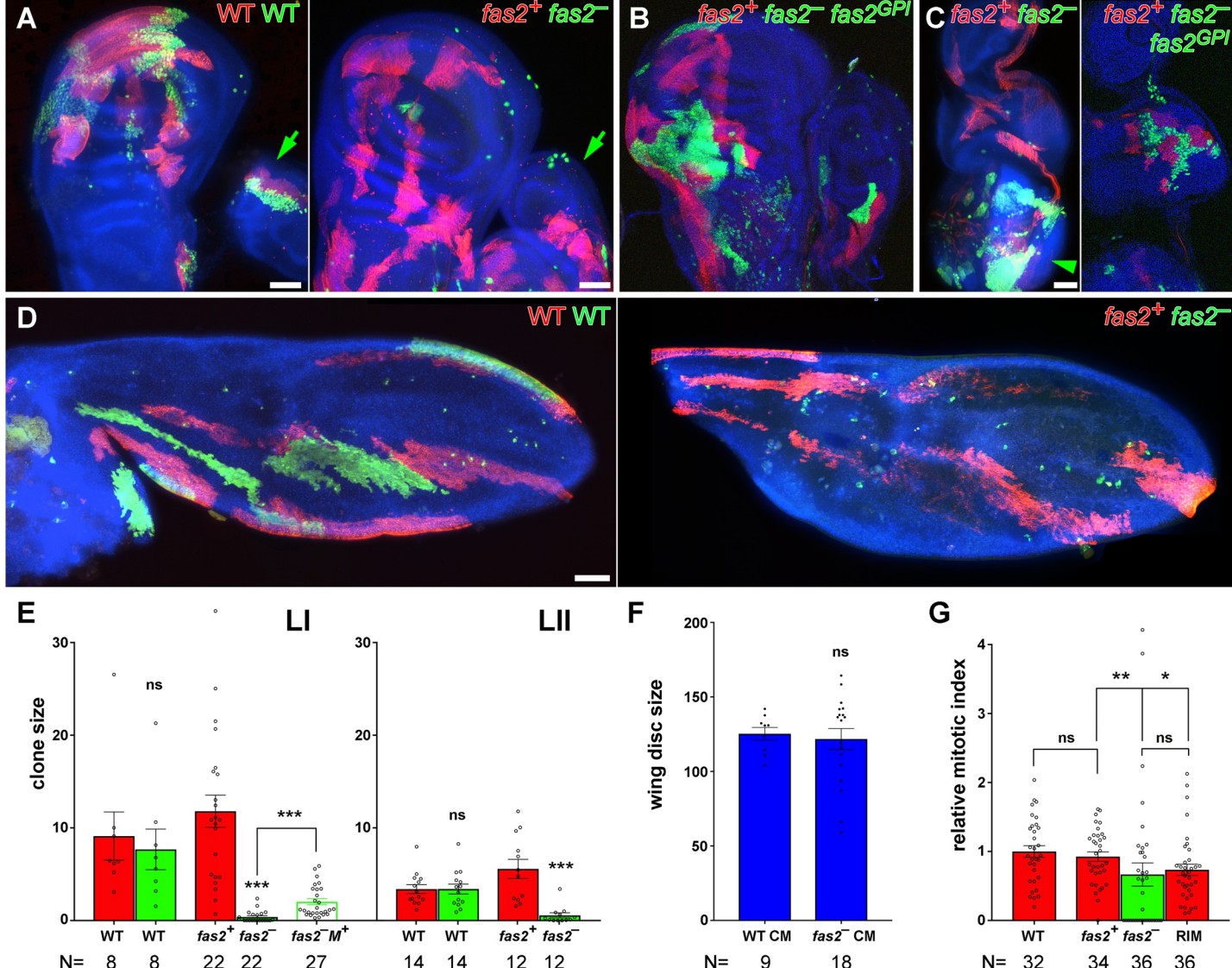

**Fig 2. Cell autonomous requirement of Fas2 for cell proliferation in imaginal discs.** (A) Left, coupled-MARCM analysis in WT. The twin clones are labeled with GFP (WT, green) and mtdTomato (WT, red), and were induced in $1^{st}$ instar larva. The green arrow points to twin clones in an adjacent leg disc. Right, coupled-MARCM analysis of the *fas2* null condition (*fas2^{eB112}*). Wing disc *fas2^−* null clones (labeled with GFP) induced in $1^{st}$ instar larva are missing or extremely small compared with their control twins (labeled with mtdTomato). Leg disc *fas2^−* clones are also missing or reduced in size compared to WT (green arrow). Bar: 50 μm. (B) Coupled-MARCM analysis of *fas2* null clones rescued by the expression of *UAS-fas2^{GPI}* isoform (GFP, green). (C) In the eye-antenna imaginal disc, *fas2^−* clones display the same behavior than in the wing disc (left), and they are also rescued by the expression of the GPI-linked isoform of Fas2 (right). Note that the *fas2^−* clones have a more normal size in the brain (arrowhead). Bar: 50 μm. (D) Left, coupled-MARCM WT twin clones in pupal wings induced in $1^{st}$ instar larva. Right, coupled-MARCM analysis of the *fas2^−* null condition in pupal wings. Fas2-deficient clones (*fas2^−* GFP) induced in $1^{st}$ instar larva are either missing or extremely small compared with their control twins (*fas2^+* mtdTomato). Bar: 50 μm. (E) Quantitative analysis of coupled-MARCM clones in the wing disc. Wing imaginal disc *fas2^−* null clones (*fas2^−* GFP, green bars) induced in $1^{st}$ (LI) and $2^{nd}$ instar larva (LII) are extremely small compared with either their internal control twins (*fas2^+* mtdTomato, red bars) or with coupled-MARCM WT twin clones (WT GFP, green bars, and WT mtdTomato, red bars). *fas2^−* *Minute^+* clones (*fas2^−* $M^+$) growing in a *Minute^−* heterozygous background display phenotypic suppression. Clone size in μm$^2$/10$^3$. N is number of clones. (F) Wing discs harboring coupled-MARCM *fas2^−* null clones (*fas2^−* CM) induced in $1^{st}$ or $2^{nd}$ instar larva have the same size than control coupled-MARCM (WT CM) discs. N is number of wing imaginal discs. (G) Comparison of the mitotic index in *fas2^{eB112}* null clones, their coupled-MARCM control twins and their contacting normal neighbor cells. The twin control clones (*fas2^+*) display the same mitotic index than WT clones. *fas2^{eB112}* null clones and the rim of normal cells around the clone show a significant reduction in the mitotic index relative to the WT value. N is number of clones.

right), indicating that *fas2^{RNAi}*-expressing cells had an enhanced activity in the JNK signaling pathway. In addition, the corresponding reduction in half of the normal dose of Puc in *fas2^{RNAi}* cells, which should cause an even higher increase in JNK activity, now produced

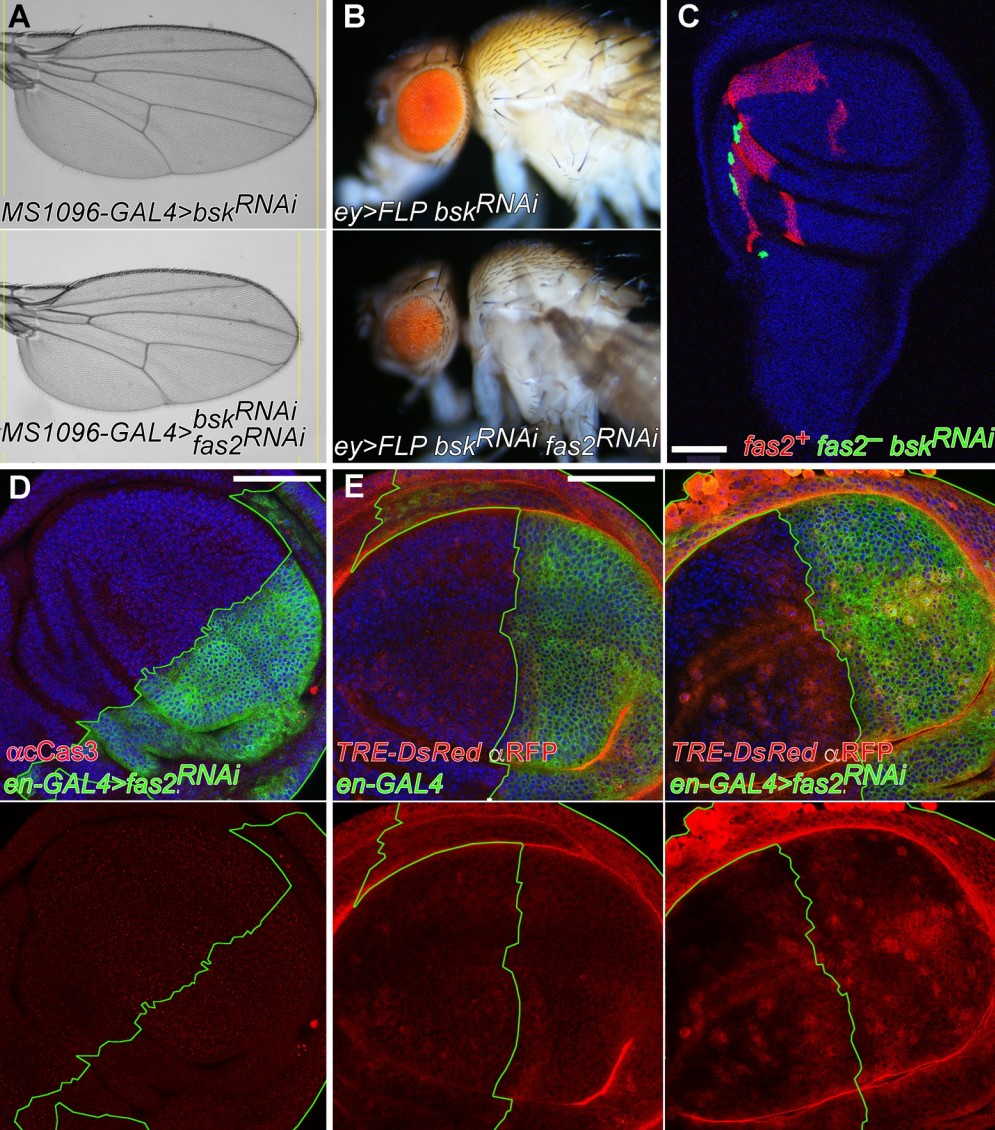

**Fig 3. Analysis of JNK signaling in *fas2* LOF conditions.** (A) Blockade of the JNK signaling pathway (using *UAS-bsk^RNAi* #32977) does not rescue the *fas2* LOF phenotype caused by *UAS-fas2^RNAi* (#34084) expression driven by *MS1096-GAL4* (double heterozygous females) (compare also with Fig 1C). (B) Expression of *UAS-bsk*^RNAi (#57035) ameliorates but does not rescue the reduced eye phenotype produced by expression of *UAS-fas2*^RNAi (#34084) in *ey*-driven FLP-OUT (*ey>FLP*) individuals (compare also with Fig 1D). (C) Coupled-MARCM *fas2^eB112* null clones with a blockade of JNK function (expressing *UAS-bsk^RNAi*, #32977) (green, GFP) display the typical reduced size compared to their normal twin controls (mtdTomato). Bar: 50 μm. (D) *en-GAL4/+; UAS-fas2^RNAi#34084/+* wing discs do not show obvious signs of apoptosis using an anti-cleaved Caspase3 antibody (red). Bar: 50 μm. (E) Inhibition of Fas2 expression in the posterior compartment of the wing disc causes a non-cell autonomous increase in JNK pathway activity. Left, the reporter of JNK activity *TRE-DsRed* (anti-RFP, red) shows a weak expression in either the posterior or anterior compartments of control wing discs. Right, the expression of *fas2^RNAi* (#34084) in the posterior compartment of the wing disc (driven by heterozygous *en-GAL4*) causes an increase of *TRE-DsRed* expression that also extends into the anterior compartment. Bar: 50 μm.

widespread apoptosis and strong alterations in the Anterior-Posterior compartment border (Fig 4B). The same genotype displayed non-cell autonomous increased expression of phosphorylated-JNK (Fig 4C).

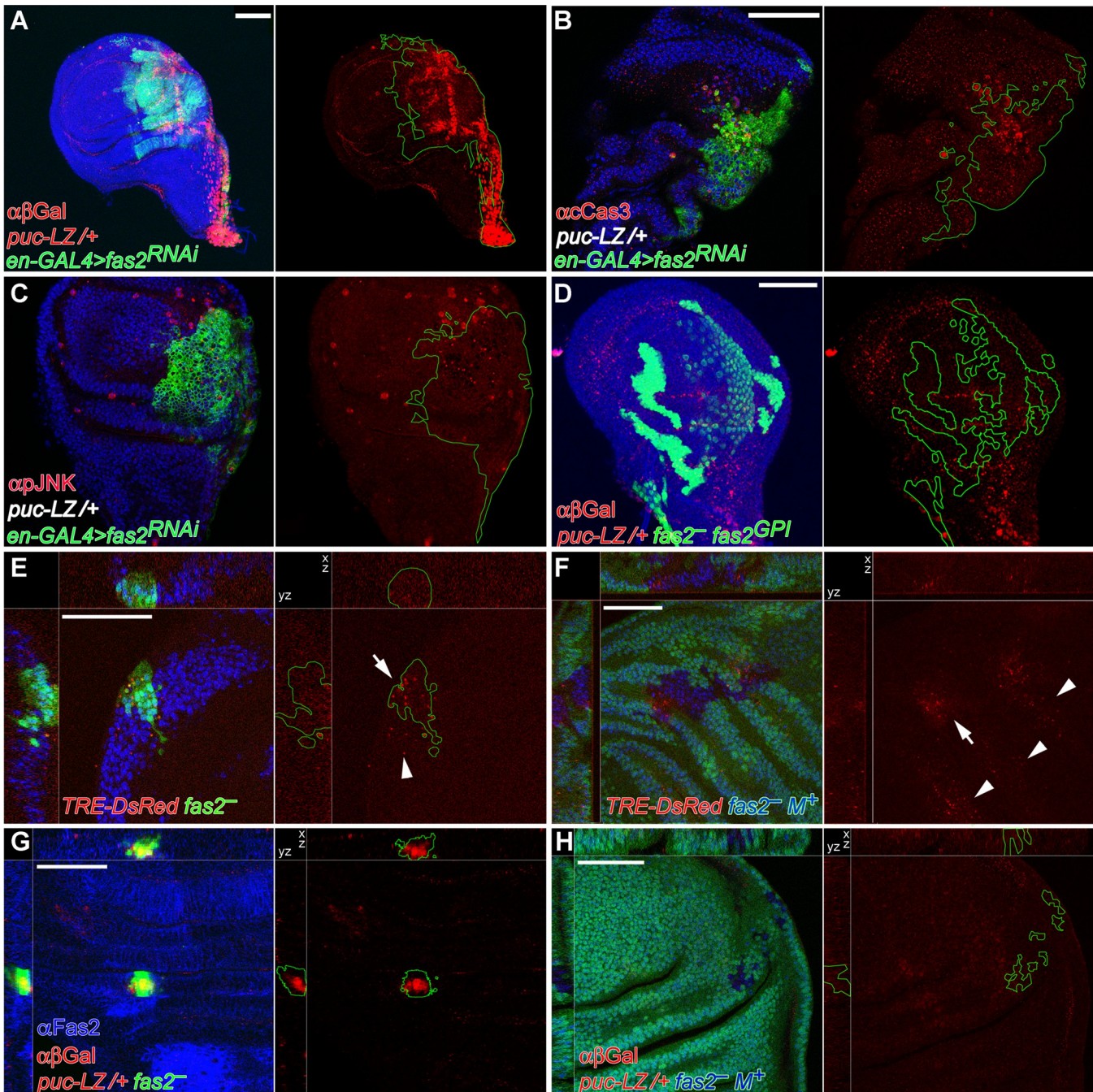

**Fig 4. Activation of JNK signaling in Fas2-deficient imaginal discs is modulated by cell competition.** (A) Posterior compartments deficient for Fas2 in *en-GAL4/+; UAS-fas2<sup>RNAi#34084</sup>/puc-LacZ* individuals display a strong expression of the reporter, indicating a strong enhancement of JNK activity. Bar: 50 μm. (B) Posterior compartments deficient for Fas2 in *en-GAL4/+; UAS-fas2<sup>RNAi#34084</sup>/puc-LacZ* individuals display widespread apoptosis (staining with anti-cleaved Caspase 3 antibody) and the breakup of the compartment border. Note that the background genotype of the whole individual is heterozygous *puc-LacZ* (*puc-LZ/+* in white). (C) The same genotype shows derepression of activated-JNK (staining with an anti-phospho-JNK antibody). Note that phosphorylated-JNK occurs non-cell autonomously in the anterior compartment as well as in the posterior compartment. The background genotype of the whole individual is heterozygous *puc-LacZ* (*puc-LZ/+* in white). (D) The Fas2<sup>GPI</sup> isoform rescues growth in MARCM *fas2<sup>eB112</sup>; UAS-fas2<sup>GPI</sup>/+; puc-LacZ/+* cell clones, which display little over-expression of *puc-LacZ*. Clones induced in 1<sup>st</sup> instar larva. (E) MARCM *fas2<sup>eB112</sup>; TRE-DsRed/+* cell clones display expression of the reporter (arrow) as well as a weaker non-cell autonomous expression (arrowhead), indicating derepression of the JNK signaling pathway. Clone induced in 2<sup>nd</sup> instar larva. (F) Slowing the proliferation rate of the cellular background (labeled with *Ubi-GFP*, using the Minute technique) reduces JNK derepression in the *fas2<sup>eB112</sup> M<sup>+</sup>* cell clones, as indicated by the expression of the reporter *TRE-DsRed* in only a few cells of the clones (arrow). Arrowheads point to non-cell autonomous expression of *TRE-DsRed* in the *Minute/+* heterozygous background. Clone induced in 2<sup>nd</sup> instar larva. (G) Compared with the *fas2<sup>eB112</sup>* rescued clones (in D), *fas2<sup>eB112</sup>; puc-LacZ/+* MARCM cell clones are extremely small and display a very strong expression of the JNK pathway reporter. Clone induced

in 1$^{st}$ instar larva. (H) Reducing the proliferation rate in the cellular background (labeled with *Ubi-GFP*, using the Minute technique) abolishes reporter expression in the *fas2$^{eB112}$M$^+$; puc-LacZ/+* cell clones (devoid of GFP expression), indicating a strong reduction in the activity of the JNK pathway. Note that the simultaneous deficit of Fas2 and half of the normal dosage of *puc* prevents these cell clones to grow as large as *fas2$^{eB112}$ M$^+$* clones with a normal dosage of *puc*. Clone induced in 1$^{st}$ instar larva.

Over-expression of the *TRE-DsRed* reporter was also found in the *fas2$^-$* cells of MARCM Fas2-deficient clones (Figs 4E and S2B, left). To determine if the JNK increased activity in the Fas2-deficient clones reflected a direct function of Fas2 on controlling this signaling pathway or indirectly resulted from the cell competition process, we analyzed *TRE-DsRed* expression in *fas2$^-$Minute$^+$* clones. The competitive advantage introduced by the *Minute$^+$* normal condition in the *fas2$^-$* clones growing in a *Minute* heterozygous background caused a suppression in *TRE-DsRed* expression (Figs 4F and S2B, left), indicating a concomitant reduction in JNK pathway over-expression. In addition, *fas2$^-$* cell clones displayed a strong expression of *puc-LacZ* (Figs 4G and S2B, right). This *puc-LacZ* expression, apoptosis and clone growth was reverted to normal by the simultaneous expression of the Fas2$^{GPI}$ isoform in the clones (Fig 4D), showing that the extracellular part of Fas2 is sufficient to support the function of Fas2 on cell proliferation in imaginal discs and revert JNK activation. Expression of *puc-LacZ* was abolished when *fas2$^-$ M$^+$* cell clones were induced in a *Minute* heterozygous background (Figs 4H and S2B, right), in addition these *fas2$^-$ M$^+$ puc-LacZ/+* clones displayed a small size. Since Puc is required for proliferation/survival in imaginal discs [36] and the *puc-LacZ* insertion causes a mutation in the *puc* gene, the combined data suggest that Puc expression is critical to balance JNK derepression to allow for Fas2-deficient cell survival and growth (see below).

## EGFR mediates the cell autonomous function of Fas2 in epithelial cell proliferation

During axon growth Fas2 promotes EGFR function [6]. In contrast, during retinal differentiation *fas2* LOF conditions cause a derepression of EGFR function [18]. Hypomorphic combinations of *fas2* display an adult phenotype reminiscent of *Egfr torpedo* (*Egfr$^t$*) alleles (S3A Fig). We analyzed the level of activated-MAPK expression (di-phosphoERK, a reporter of EGFR activity) in *eyeless*-driven (*ey>FLP*) *fas2$^{RNAi}$* FLP-OUT imaginal discs. These Fas2-deficient eye imaginal discs displayed a low level of activated-MAPK expression compared to their control siblings, or tissue in the same individual without *ey* expression (Fig 5A; quantified in S3B Fig), revealing reduced EGFR activity. These results show that Fas2 promotes EGFR activity during imaginal disc growth. Indeed, a 50% reduction of the *Egfr* dosage enhanced the phenotype of *fas2* LOF conditions (Figs 5B and S3C), showing that Fas2 function is highly sensitive to the dosage level of EGFR. Moreover, double mutant combinations between *Egfr* and *fas2* strong LOF conditions displayed an epistatic behavior (Fig 5C). The results are strongly consistent with a positive interaction of Fas2 and EGFR in the same signaling pathway during imaginal disc growth. Therefore, Fas2 function during imaginal disc growth seems just opposite to its reported function during retinal differentiation.

To further analyze the specificity of the interaction between Fas2 and the EGFR, and to identify other effectors that may mediate the Fas2 function in imaginal discs, we studied the capacity of different GOF conditions for proteins involved in growth signaling to suppress the phenotype of *fas2$^-$* clones (S3D Fig). Activated-EGFR (*UAS-λEgfr*) rescued the size of *fas2$^-$* clones in imaginal discs, pupa and adult (Fig 5D and 5F). Expression of activated-EGFR in *fas2$^-$* cell clones caused a significant increase in the mitotic index within the clone, while it remained low in the rim of Fas2-normal cells surrounding the clone (Fig 5G, compare to Fig 2G). This result is strongly consistent with the idea that Fas2 homophilic binding is required

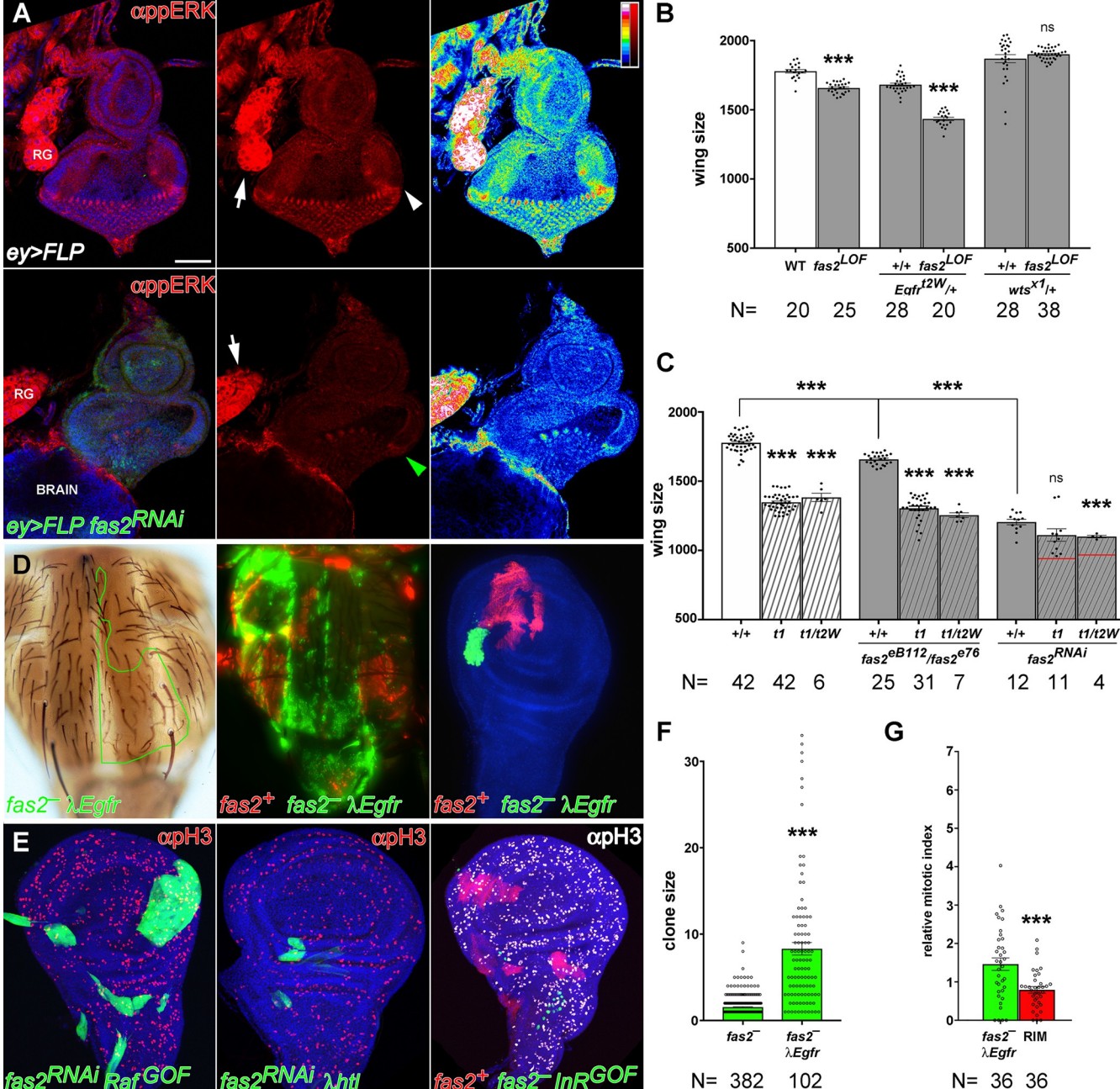

**Fig 5. The EGFR signaling pathway mediates the Fas2 cell autonomous function to promote imaginal disc growth.** (A) Eye imaginal discs deficient for Fas2 display reduced levels of activated-MAPK (anti-ppERK). Top panels, strong expression of activated-MAPK is evident in the Morphogenetic Furrow of the eye disc (white arrowhead) and the Ring Gland (RG, arrow) of control siblings (CyO balancer siblings). Bottom panels, in contrast *ey*-driven FLP-OUT eye discs expressing *UAS-fas2^RNAi (#34084)* show a strong reduction of the ppERK signal, especially evident in the Morphogenetic Furrow (green arrowhead). Note that the Ring Gland (RG, white arrow), where *ey-FLP* is not expressed, serves as an internal control for the normal activated-MAPK signal. The right panels show a LUT color coded representation of activated-MAPK signal levels (signal quantified in S3B Fig). Bar: 50 µm. (B) The *fas2* wing phenotype is sensitive to the dosage of the *EGFR* and *wts* genes. *fas2^eB112/fas2^e76* wings show a reduction in size which is enhanced by a reduction of 50% in the dosage of *EGFR* (*Egfr^top2W74/+*). Reciprocally, the *fas2* LOF phenotype is suppressed by a reduction of 50% in the dosage of *wts* (*wts^X1/+*) (WT and *fas2^eB112/fas2^e76* as in Fig 1G). All individuals are female. Wing size is wing area in µm²/10³. N is number of individuals. (C) Functional interactions between *fas2* and *Egfr* LOF mutations. The size of *fas2^eB112/fas2^e76* and *fas2^RNAi34084* (driven by heterozygous *MS1096-GAL4/+*) adult wings (grey bars) is significantly smaller than WT (white bar). *Egfr* hypomorphic combinations (*Egfr^top1/Egfr^top1*, *t^1*, and *Egfr^top1/Egfr^top2W74*, *t^1/t^2W*) display a wing size smaller than WT and *fas2^eB112/fas2^e76* (striped white bars), and more similar to that caused by the expression of *UAS-fas2^RNAi34084*. The wing size of the double mutant combinations of *fas2^eB112/fas2^e76* with the *Egfr* allelic combinations is more similar to the *Egfr* mutants (striped grey bars), and shows a clear epistatic interaction with the stronger *fas2^RNAi34084* condition. Note that the combination with *Egfr^top1/Egfr^top2W74* still causes some enhancement, but far from the value expected for an additive effect of the fas2

and Egfr phenotypes (red lines). Controls as in Fig 1G. All individuals are female. Wing size is wing area in $\mu m^2/10^3$. N is number of individuals. (D) Rescue of the $fas2^-$ phenotype by activated-EGFR ($\lambda Egfr$) in adult (left), late pupa (middle) and the wing imaginal disc. At left, an adult $fas2^{eB112}$ $\lambda Egfr$ MARCM clone in the notum marked with *yellow* and *forked* (outlined in green) showing a rescued size. Middle, $fas2^{eB112}$ $\lambda Egfr$ coupled-MARCM clones in the notum (GFP) displaying a rescued size similar to their normal $fas2^+$ control twins (labeled with mtdTomato). Right, $fas2^{eB112}$ $\lambda Egfr$ coupled-MARCM clone and control twin (labeled with mtdTomato) in the wing disc. (E) Expression of activated-Raf ($UAS-Raf^{GOF}$), but not of activated-FGFR ($UAS-\lambda htl$) or activated-InR ($UAS-InR^{DEL}$), rescues $fas2$-deficient clone growth. Left, $2^{nd}$ instar larva FLP-OUT $UAS-fas2^{RNAi\#34084}$ $UAS-Raf^{GOF}/+$ clones. Middle, $2^{nd}$ instar larva FLP-OUT $UAS-fas2^{RNAi\#34084}$ $UAS-\lambda htl/+$ clone. Right, $2^{nd}$ instar larva coupled-MARCM $fas2^{eB112}$; $UAS-InR^{DEL}/+$ clones (GFP, absent) and their control twins (mtdTomato). A similar result was obtained with the expression of $UAS-InR^{R418P}$. (F) $y$ $fas2^{eB112}$ $f^{36a}$ MARCM $2^{nd}$ instar larva clone size in adults is rescued by activated EGFR ($UAS-\lambda Egfr/+$). Clone size is number of $y$ $f^{36a}$ chaetes. N is number of clones. (G) The mitotic index (relative to WT, as in Fig 2G) in $fas2^-$ $\lambda Egfr$ clones is highly increased, while the 2–3 cell rim of $fas2^+$ cells around the clone shows the same value than in the rim of $fas2^+$ cells around $fas2^-$ clones (compare with Fig 2G). N is number of clones.

for promoting EGFR activity. Effectors of the EGFR signaling pathway: activated-Ras ($UAS-Ras^{V12}$), activated-Raf ($UAS-Raf^{GOF}$), active-PI3K ($UAS-Dp110$) (Figs 5E, left, and S3D) and over-expression of Yki ($UAS-Yki$) (Figs 6A, left, and S3D) also produced a significant normalization of $fas2^-$ MARCM clones in the adult and imaginal discs. Therefore, the results indicate that EGFR and its effectors function downstream of Fas2 during imaginal disc growth. In contrast, activated-FGFR ($UAS-\lambda htl$), activated-Insulin receptor ($UAS-InR^{DEL}$, $UAS-InR^{R418P}$), activated-Notch ($UAS-N^{INTRA}$) or myristoylated Src ($UAS-Src^{myr}$) were unable to produce a significant effect in adult or imaginal disc clones (Figs 5E, middle and right, and S3D).

## The Hippo signaling pathway is indirectly involved in the fas2⁻ phenotype

LOF conditions for Fas2 have been shown to cause over-proliferation in the follicular epithelium of the ovary and to genetically interact with *warts* (*wts*), suggesting that the Hippo signaling pathway mediates Fas2 function in this tissue [19]. We tested the interaction of *fas2* with *wts* during imaginal disc development. A reduction in half of the *wts* gene dosage rescued the wing size phenotype of the hypomorphic $fas2^{eB112}/fas2^{e76}$ combination (Fig 5B). To analyze the functional epistatic relationship between Fas2 and the Hippo pathway in the wing imaginal disc, we blocked the function of *expanded* (*ex*) and *wts* in $fas2^{RNAi}$ FLP-OUT clones. Ex and Wts are required to repress cell proliferation in imaginal discs and their LOF conditions cause overgrowth, therefore we expected their inhibition to be epistatic to the Fas2 phenotype and cause a rescue of *fas2* clone growth. However, neither $ex^{RNAi}$ nor $wts^{RNAi}$ expression was able to suppress the growth deficit of Fas2-deficient clones (Fig 6A middle and right), suggesting that the Hippo pathway does not directly mediate Fas2 function in the growing imaginal discs. Since the EGFR and the Hippo pathways interact to control Yki activity [37,38], which in turn feeds-back on Ex expression, we studied the expression of the Yki reporter *ex-LacZ* [39] in $fas2^-$ cells. We compared the expression of this Yki reporter between each posterior compartment and its anterior counterpart in *en-GAL4 UAS-fas2^{RNAi}* wing imaginal discs (Fig 6B; quantified in S4 Fig). We observed a significant increase of *ex-LacZ* expression, suggesting that Yki is over-activated in the *fas2* LOF condition. It is known that the JNK and Hippo signaling pathways interact [30,31], therefore we tested if the enhanced expression of the reporter in the $fas2^{RNAi}$ condition may be a consequence of the increased activity in the JNK pathway (results above). Suppression of JNK activity in the Fas2-deficient posterior compartment reverted the increase in *ex-LacZ* expression (Fig 6B right; quantified in S4 Fig). These results together with the previous ones strongly suggest that the indirect activation of the JNK pathway in Fas2-deficient cells promotes Yki activity as a compensatory mechanism for the EGFR-dependent growth deficit.

## EGFR and Yki activity promote Fas2 expression

Fas2 and EGFR are co-expressed and colocalize in all cells of the growing imaginal discs (S5A Fig). EGFR LOF conditions have been shown to cause a reduction of Fas2 expression in

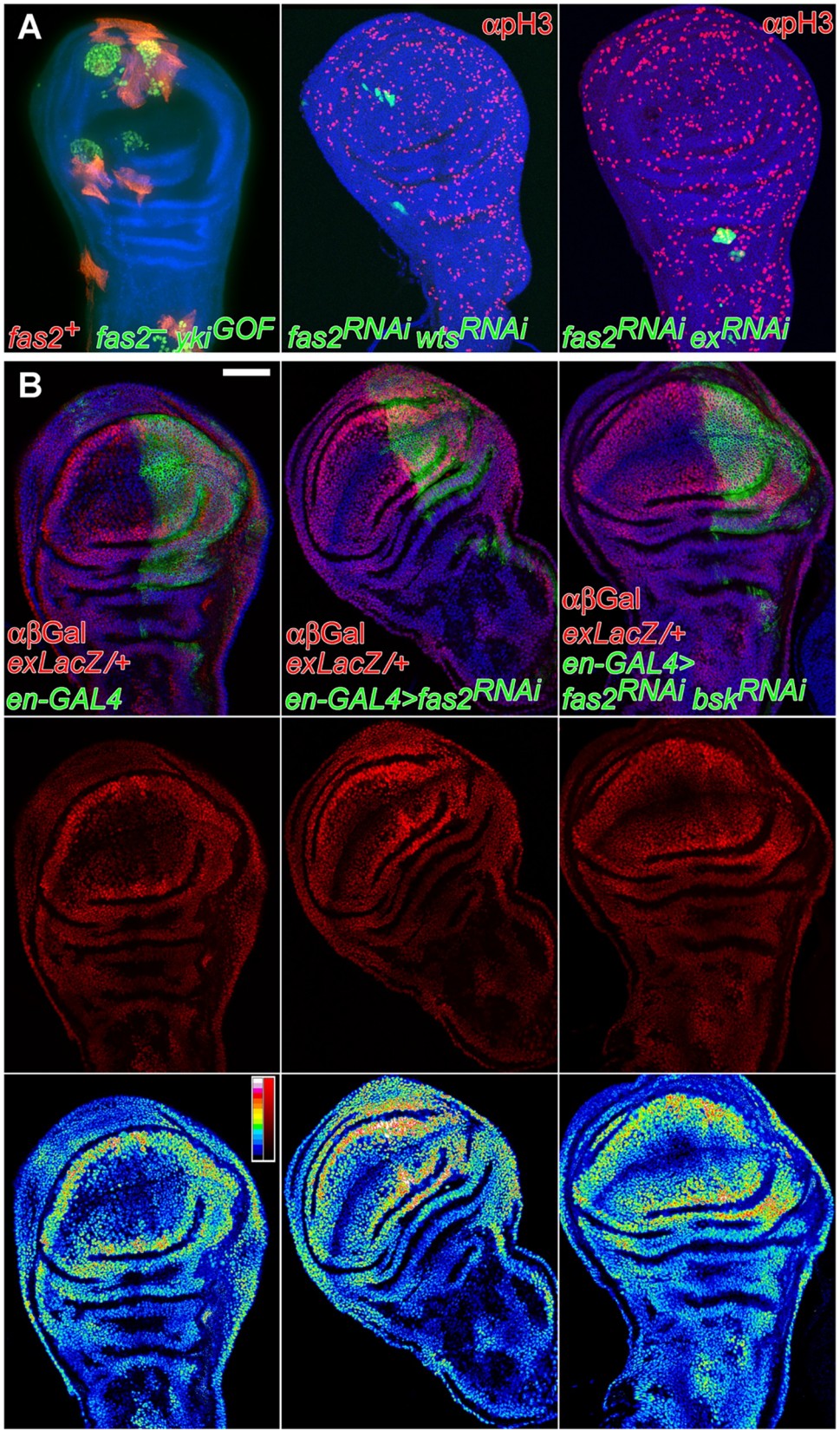

**Fig 6. The Hippo pathway indirectly interacts with Fas2 during imaginal disc growth.** (A) Left, rescue of the *fas2⁻* clone phenotype by Yki over-expression. Coupled-MARCM *fas2^eB112* clones over-expressing Yki (*fas2^eB112; UAS-yki/+*, labeled with GFP) and their *fas2⁺* control twins (labeled with mtdTomato) in the wing disc. Middle and right panels, inhibition of Wts or Ex expression does not modify the loss of growth in Fas2-deficient cell clones. Middle, 2^nd instar larva *UAS-fas2^RNAi#34084 UAS-wts^RNAi#37023* FLP-OUT clones (labeled with GFP). Right, 1^st instar larva *UAS-fas2^RNAi#34084 UAS-ex^RNAi#34968* FLP-OUT clone (labeled with GFP). (B) Reduction of Fas2 function by the expression of *UAS-fas2^RNAi* (#34084) under the control of the *en-GAL4* driver causes a JNK-dependent over-expression of the *ex-LacZ* reporter. Left panels show a control *en-GAL4/ex-LacZ* wing disc. Middle panels show an *en-GAL4/ex-LacZ; UAS-fas2^RNAi34084/+* wing disc. Note the higher β-Gal expression in the P compartment extending into the A compartment (see bottom panels for a color coded LUT representation of β-Gal signal intensity to compare levels). Right panels show an *en-GAL4/ex-LacZ; UAS-fas2^RNAi34084/UAS-bsk^RNAi#32977*. Note the reversion in the β-Gal expression from the *ex-LacZ* reporter caused by the inhibition of JNK (signal intensities quantified in S4 Fig). Bottom panels show a color coded LUT representation of β-Gal signal intensity to compare levels. Bar: 50 μm.

imaginal discs [18]. We confirmed this result (S5B Fig). In addition, we found that cell clones expressing activated-EGFR or activated-Ras display a cell autonomous increase in the expression level of Fas2 (Fig 7A). This indicates that EGFR activity is not merely a permissive requirement for Fas2 expression in imaginal discs, but an instructive signal. Since EGFR function controls Yki activity [38], we tested if Yki over-expression could also cause a change in Fas2 expression in wing imaginal disc cell clones. We also found a strong enhancement of Fas2 expression in clones over-expressing Yki (Fig 7B). It has been shown that EGFR over-activation and JNK activity can interact to promote Yki expression (see [40]). To see if Fas2 over-expression in the EGFR over-activation condition requires the function of JNK, we generated *λEgfr bsk^RNAi* cell clones in a Fas2::GFP protein trap background. The suppression of JNK activity in these clones did not reduce the over-expression of Fas2 (S6C and S6D Fig), demonstrating that expression of Fas2 is directly controlled by the EGFR signaling pathway. These results together with the previous ones point to the existence of a cell autonomous self-stimulating feedback loop between EGFR activity and Fas2 expression during imaginal disc growth (Fig 7E top).

## Fas2 physically interacts with EGFR

Our genetic data suggested that Fas2 and EGFR might physically interact at the plasma membrane. To explore this possibility we tagged the Fas2^TRM protein isoform with a V5 epitope, and co-expressed it together with *Drosophila* EGFR (dEGFR) in HEK293 cells. Then we used an anti-V5 mouse monoclonal antibody to immunoprecipitate Fas2. A very strong anti-dEGFR immunoreactivity was pulled down from cells expressing Fas2 and dEGFR, while only background immunoreactivity was pulled down from cells that expressed dEGFR but not Fas2 (Fig 7C). To confirm the interaction, we performed the reverse experiment. Immunoprecipitation with anti-dEGFR antibody was able to pull down Fas2 in cells that co-expressed dEGFR. However, no V5 immunoreactivity was detected when the dEGFR was not co-expressed in the cells (Fig 7D). Together, our results demonstrate that, in the HEK293 heterologous system, Fas2 physically interacts with *Drosophila* EGFR.

## Discussion

Fas2 is the Drosophila ortholog of the vertebrate homophilic cell adhesion molecule NCAM. In addition to its role in recognition/adhesion, Fas2 has been shown to promote EGFR function during axon growth [6]. Conversely, Fas2 has been also reported to function as an EGFR repressor during retinal differentiation [18]. Here we have studied the requirement for Fas2 during imaginal disc growth. Fas2 is dynamically expressed by all epithelial cells in imaginal discs. It is maximally expressed in pre-differentiating and differentiating structures, like veins

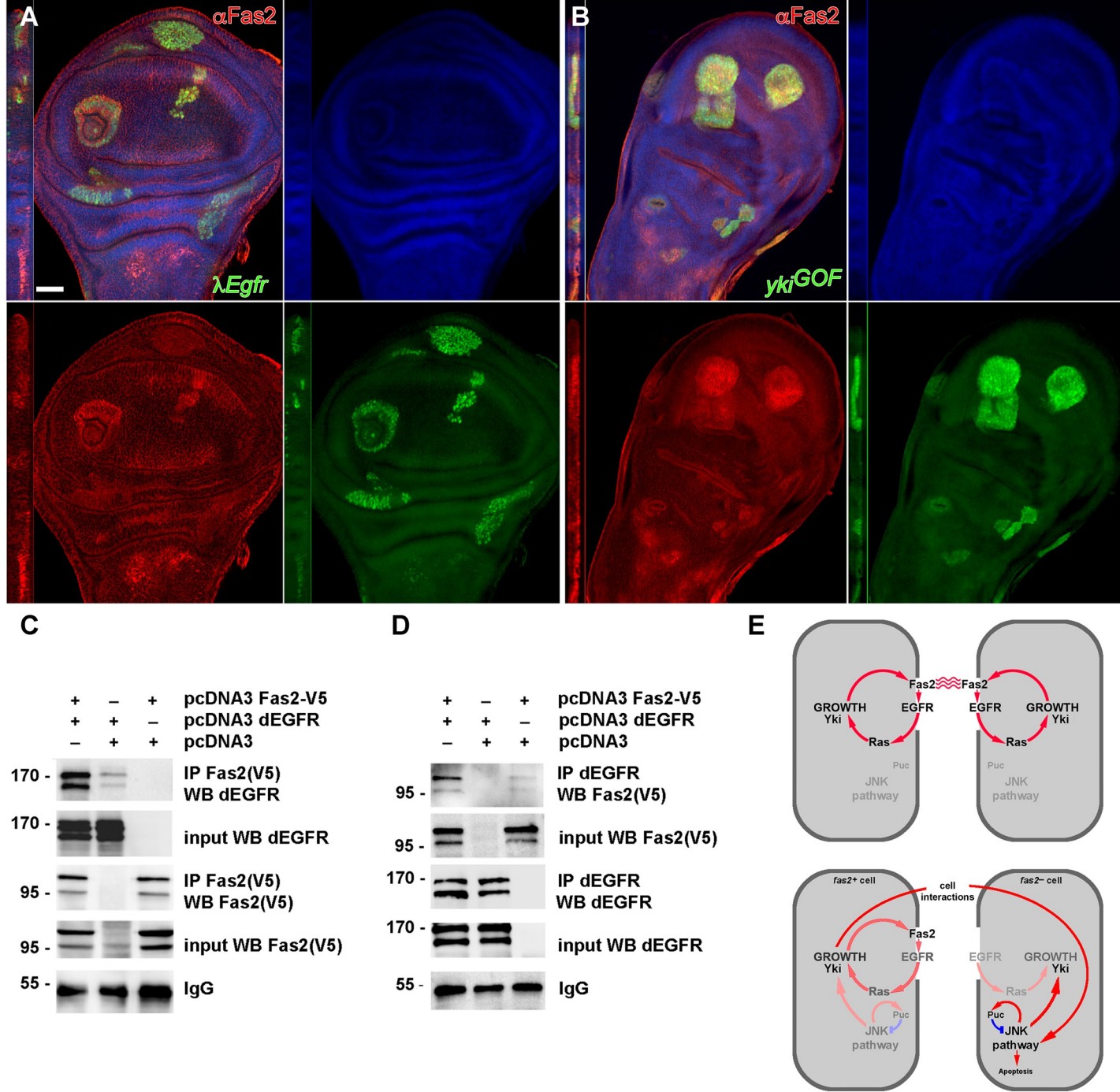

**Fig 7. EGFR physically binds Fas2 and promotes its expression in imaginal discs.** (A) Activated-EGFR MARCM clones (λEgfr, labeled with GFP) show an increased cell autonomous expression of Fas2 (stained with the anti-Fas2 antibody MAb 1D4, red channel) in the wing imaginal disc. Note that Fas2 over-expression exactly matches the contour of the clones. Clones induced in 2ⁿᵈ instar larva. Bar: 50 μm. (B) MARCM clones over-expressing Yki (ykiGOF, labeled with GFP) display increased expression of Fas2 (in red, 1D4 antibody). Clones induced in 2ⁿᵈ instar larva. (C) Fas2 co-immunoprecipitates with dEGFR. Two plasmids containing the cDNA encoding for Fas2-V5 and dEGFR were transiently co-transfected into HEK293 cells (first lane). 24h later, cells were homogenized and Fas2 immunoprecipitated (IP) with the mouse anti-V5 antibody. dEGFR was strongly pulled down from these cells, while only background levels were obtained from control cells transfected with dEGFR but without Fas2 (center lane). dEGFR expression was similar in both extracts (input). The immunoblot with anti-V5 showed that the Fas2 protein was correctly expressed (input) and immunoprecipitated. IgG shows a similar load of immunoprecipitate. (D) Reverse co-immunoprecipitation. The IP with anti-dEGFR antibody pulls down Fas2 in cells co-transfected with dEGFR and Fas2 (left lane), but not in cells transfected with Fas2 alone (right lane). Fas2 expression was similar in both extracts (input). Immunoblot with anti-dEGFR showed that dEGFR was correctly expressed (input) and immunoprecipitated. IgG shows a similar load of immunoprecipitate. (E) Summary of functional interactions between Fas2, EGFR and JNK during imaginal disc growth. Top, during larval development, Fas2 signals

through the EGFR in imaginal discs to stimulate cell proliferation and its own expression in a cell autonomous manner. The activity in the JNK pathway remains minimal. Bottom, a cell deficient for Fas2 lacks the corresponding homophilic interactions with their normal neighbors. In such an scenario, EGFR activity is low, and also somewhat reduced in the Fas2-normal contacting cells (note that these still keep their normal Fas2 homophilic binding with other Fas2-normal cells). In the absence of Fas2 homophilic binding, cell to cell interactions promote the activity of JNK producing an increase in Yki expression. The negative feed-back on JNK provided by Puc is critical to prevent shifting the balance in JNK expression towards apoptosis.

and sensory organ precursor cells in the wing disc, and the Morphogenetic Furrow and the differentiating retina in the eye disc. Our LOF analyses reveal that Fas2 is required for cell proliferation and that imaginal disc growth depends on the level of Fas2 function. Fas2-deficient cells in whole organs can survive and differentiate epidermis in adults, however Fas2 null cell clones grow less than their normal control twins (which grow as WT clones) and can be out-competed by the normal neighbor cells. Fas2-deficient surviving cell clones show a reduced mitotic index compared to their Fas2-expressing control twins or WT clones, in addition the normal cells contacting the Fas2-deficient clones also show a reduced mitotic index. This behavior most likely reflects a dependence of proliferation on the Fas2 homophilic interaction between cells, rather than a requirement for the simple presence of the protein. The *fas2⁻* clone growth deficit is significantly suppressed when the rest of the disc has a genetic background that slows the general proliferation rate (a condition generated using the Minute technique), and it is lightly corrected when inhibiting apoptosis within the clone. The Fas2 cellular requirement for imaginal disc growth is corrected by expressing the Fas2$^{GPI}$ isoform in the Fas2-deficient cells, showing that the extra-cytoplasmic part of Fas2 is sufficient to support its function during proliferation.

The loss of growth in Fas2-deficient clones correlates with an over-activation of the JNK signaling pathway, as revealed by the expression of the reporters *TRE-DsRed* and *puc-LacZ* in cell clones or whole compartments deficient for Fas2. JNK signaling is involved in imaginal disc and stem cell homeostasis [41,42]. We found that JNK over-activation is not the main cause of the reduced growth phenotype of Fas2-deficient cell clones, since blocking JNK activity by the expression of *bsk* RNAi does not correct their growth. The increased activity of the JNK signaling pathway depends on the different growth rate between the Fas2-deficent clones and their Fas2-normal background, as revealed by its suppression in the *fas2⁻ Minute⁺* cell clones. In addition, we found that a normal dosage of the JNK feedback repressor *puc* in the genetic background is critical for the survival of Fas2-deficient cells as well as for the growth of the *fas2⁻M⁺* clones, since a reduction in 50% of *puc* dosage causes widespread cell death of *fas2⁻* cells and impedes *fas2⁻ M⁺* clones to grow as much as in a normal-*puc* genetic background. Thus, the expression level of Puc plays a pivotal role in both Fas2-deficient clone survival and growth (see Fig 7E for a summary of the functional interactions between Fas2, EGFR and JNK).

Fas2 physically binds EGFR. Interestingly, EGFR activity has been recently shown to be involved in cell competition [43,44]. We have found that the requirement for Fas2 during imaginal disc growth reflects a direct function to promote EGFR activity. Fas2-deficient eye imaginal discs show a reduced level of MAPK activation. LOF conditions of *fas2* display phenotypes reminiscent of *Egfr* LOF conditions, and the genetic interaction epistasis of *fas2* and *Egfr* LOF mutant combinations shows that Fas2 and EGFR function in the same developmental pathway during imaginal disc growth. Moreover, the cell autonomous slow growth of *fas2⁻* clones is rescued by the expression of activated-EGFR or activated forms of its downstream effectors: Raf, Ras or PI3K. Since both the Fas2$^{GPI}$ isoform and an increased EGFR activity were sufficient to rescue the growth deficit of Fas2-deficient cells, the protein interaction between Fas2 and EGFR probably involves their extracellular domains, as it happens for the functionally similar interaction between L1-CAM and EGFR [6,14]. Interestingly, the

expression of activated-FGFR, which shares most of the downstream effectors with the EGFR [45], was unable to produce any rescue in growing imaginal discs. This reveals a high degree of specificity in the molecular interactions that mediate Fas2 function in different tissues during development. It is very significant that EGFR activity in turn drives the cell autonomous expression of Fas2 in imaginal discs. LOF and GOF conditions of *Egfr* cause corresponding changes in the expression level of Fas2, revealing that this is an instructive function of EGFR. Thus, the combined results of the *fas2* LOF analysis and its genetic interactions with *Egfr* show the existence of a cell autonomous, self-stimulating, positive feedback loop between Fas2 and the EGFR signaling pathway during imaginal disc growth (Fig 7E). This auto-stimulatory loop may relate to the previously identified positive feedback amplification loop of EGFR activity required for vein formation during imaginal disc growth [22].

In addition to its interaction with the EGFR, *fas2* has been shown to interact with the Hippo pathway effector *wts* during proliferation of the follicular epithelium in the ovary [19]. Since the Hippo signaling pathway controls cell proliferation during imaginal disc development [46], we have also studied its possible function as mediator of Fas2 function in imaginal disc growth. We found that a 50% reduction of *wts* dosage can rescue the loss in wing growth associated with an hypomorphic *fas2* LOF condition. However, blocking the Hippo pathway by the expression of *ex* or *wts* RNAis does not rescue Fas2-deficient cell clones, ruling out the Hippo pathway as a direct effector of Fas2 in growing imaginal discs. On the other hand, JNK does control the Hippo pathway during imaginal disc development [31], as well as during cell competition interactions and compensatory growth [29,35,47]. Indeed, we detected an increase in the expression of the Yki reporter *ex-LacZ* in Fas2-deficient wing disc compartments. Moreover, this enhanced expression was reduced when we simultaneously inhibited the JNK pathway with $bsk^{RNAi}$. Since JNK has a function to promote cell growth [48], the enhanced expression of Yki in *fas2⁻* cells may represent a compensatory response to the loss of EGFR activity (see Fig 7E for a summary of interactions between Fas2, EGFR and JNK). In addition, we also found that Yki over-expression can rescue Fas2-deficient clone growth. Interestingly, *Minute* slow growing cells depend on Yki for growth and survival during imaginal disc development [49]. Since Yki has been shown to be controlled by both EGFR and JNK activity [30,38], its involvement in mediating the *fas2⁻* phenotype most probably reflects its dependence on both, a direct effect of Fas2 deficit on EGFR activity as well as an indirect compensatory effect via JNK action on the Hippo pathway.

The results of the analysis of Fas2 function in growing imaginal discs sharply contrast with those reported in the differentiating retina [18]. While Fas2 promotes EGFR function in the growing imaginal disc epithelium, it represses EGFR function in the eye imaginal disc differentiating cells. This suggests that Fas2 can be both a cell autonomous EGFR activator at low and moderate expression levels (like those during imaginal disc growth) and an EGFR repressor at high levels of expression (like those in differentiating cells). In accordance with this idea, it is interesting to note that during synapse growth a 50% expression level of Fas2 causes the largest sizes, while low and high expression levels cause a reduction of growth [50]. Interestingly, the presence of the auto-stimulatory loop between Fas2 and EGFR suggests that Fas2 expression could increase with time and the two functional facets of Fas2 may operate in a concerted manner during imaginal disc growth to set a threshold to stop EGFR-promoted growth.

## Materials and methods

### Drosophila strains and genetic crosses

A description of the different *Drosophila* genes and mutations can be found at FlyBase, *www. flybase.org*. *Drosophila* stocks were obtained from the Bloomington Stock Center and Vienna

Drosophila Research Center. The Fas2::GFP line was a gift from Dr. Christian Klambt. The different strains used in the mosaic analyses can be found in the supplementary Materials and Methods (S1 Text). Clones were induced by 1-hour heat-shock at 37˚ C, for MARCM and coupled-MARCM analyses, and a 10 minutes heat-shock, for FLP-OUT clones. This yielded an average frequency of 1 clone per wing disc and 1 notum clone every 5 adult flies for clones generated during $1^{st}$ instar larva. Clones were induced during $1^{st}$ (24–48 hours after egg laying, AEL) or $2^{nd}$ (48–72 hours AEL) larval instars. All genetic crosses were maintained at 25±1˚C in non-crowded conditions. To obtain *fas2⁻ Minute⁺ puc-LacZ* clones, we crossed *fas2^{eB112} FRT19A; hs-fas2^{TRM}/puc-LacZ* males with *Ubi-GFP M(1)O^{sp} FRT19A/FM7; hs-FLP* females and induced clones in $1^{st}$ and $2^{nd}$ instar larva. *puc-LacZ/+* imaginal discs were identified by the normal expression of *puc* in the peripodial membrane at the base of the wing disc. To obtain *fas2⁻ Minute⁺ TRE-DsRed* clones, we crossed *fas2^{eB112} FRT18A; hs-fas2^{TRM}/hs-FLP* males with *Ubi-GFP M(1)O^{sp} FRT18A/FM7; TRE-DsRed* females and induced clones in $1^{st}$ and $2^{nd}$ instar larva.

## Fas2 and EGFR co-immunoprecipitation

The pcDNA3 construct encoding *Drosophila* EGFR (*DER 2*) was a gift of Prof. E. Schejter (Dept. of Molecular Genetics, The Weizmann Institute of Science, Rehovot, Israel). The construct *pOT2-Fas2* was obtained from the *Drosophila* Genomics Resource Center. The cDNA encoding for *Drosophila fas2* was subcloned into the *pcDNA3.2/V5/TOPO* vector (Invitrogen). Human embryonic kidney cells (HEK293) were cultured in Dubecco´s modified Eagle´s medium (DMEM) containing 10% of fetal bovine serum. Cells were plated at sub-confluence, and twenty hours later transfected with the indicated plasmids using lipofectamine 2000 following the manufacturer recommendations. To improve Fas2 expression, cells were incubated for 12h with 2 μM MG132. 24h post-transfection cells were lysed (lysis buffer: 50mM Tris, 150mM NaCl, 2mM EDTA, protease inhibitor cocktail -Complete Mini, Roche-, 0.5% Triton X-100) and incubated 1h on ice. Cell lysate was centrifuged 10min at 4˚C, and an aliquot of the supernatant was kept aside on ice ("input"). Protein A-Sepharose beads (GE Healthcare) were loaded with rabbit anti-V5 tag ChIP grade (Abcam; ab9116) or mouse anti-*Drosophila* EGFR (C-273) antibody (Abcam; ab49966) for 1h at room temperature and washed 3 times with PBS. Cell lysate supernatant was mixed with antibody-loaded beads, and incubated 3h on ice, with mild shaking. Beads were washed 4 times with ice-cold PBS, resuspended in SDS sample buffer, boiled 5min, and submitted to SDS-PAGE in a 7% acrylamide gel. The proteins were transferred to nitrocellulose membrane (Protran, Whatman GmbH). The membrane was blocked with 5% BSA in TBS containing 0.1% Tween and incubated with the indicated antibody in blocking buffer overnight at 4˚C. The membrane was then washed three times with TBS containing 0.1% Tween, and the corresponding secondary antibody (horseradish peroxidase-conjugated: SIGMA anti rabbit IgG-HRP -A9169- and anti-mouse IgG-HRP -A4416-) was applied at 1:2000 dilution in TBS containing 0.1% Tween for 2h at room temperature. Immunoreactivity was detected using ECL Plus detection reagent (GE Healthcare).

## Immunohistochemistry and data acquisition

We used the following primary antibodies: anti-Futsch MAb 22C10 for PNS neurons, anti-Fas2 MAb 1D4 and anti-βgal MAb 40–1 (DSHB, University of Iowa); anti-βgal rabbit serum (Capel); rabbit anti-cleaved Caspase3 and anti-phospho-Histone H3B (Cell Signaling Technology), and anti-activated MAPK (Sigma). Staining protocols were standard according to antibody specifications. For surface measurements, images of adult nota and wings were acquired at 40X, pupal wing images at 100X, and imaginal discs at 200X. Surface area measurements

were taken in pixels (pxs) and expressed as $\mu m^2$ according with the calibration of microscope objective and digital camera. Measured wing area corresponded to the region from the alula to the tip of the wing. Images were obtained in a Leica DM-SL confocal or a Leica Thunder microscope, or in a Nikon Eclipse i80 microscope/Optigrid Structured Light System. Images were processed using Volocity 4.1–6.1 software (Improvision Ltd, Perkin-Elmer).

## Statistical analysis

To calculate the mitotic index in the clone, its rim and the rest of the imaginal disc, we outlined each GFP clone with the Photoshop selection tool. To obtain the rim area of normal tissue corresponding to some 2–3 cell diameters around the GFP clone, we extended the selection area of the GFP signal by 25 pixels out of the clone border and removed the area of the clone itself. The area between this rim and the border of the disc defined the rest of the tissue. We counted the number of pH3 positive spots in each area and divided this number by the area in $\mu m^2$. All values were then expressed as ratio to the mitosis/area value of the WT control clones. We used paired Student's *t*-test (two-tailed) when possible, as it is the most powerful and stringent test to compare differences between clone twins and between each clone, its rim and its imaginal disc background. In other cases, to compare values between different populations, we used unpaired Student's *t*-test (two-tailed), Student's *t*-test with Welch's correction, or the Mann-Whitney test according with variances and distribution of values. Error bars in Figures are SEM. Significance value: * $P<0.05$, ** $P<0.01$ and *** $P<0.001$. Statistical software was Prism 7.0d, GraphPad Software, San Diego California USA, *www.graphpad.com*.

## Supporting information

**S1 Fig. Fas2 requirement in imaginal discs.** (A) Gynandromorph (*y fas2^{eB112}/R(1)2*) displaying a Fas2-deficient right side (*y fas2^{eB112}*, labeled with the *yellow* marker, green outline). (B) Coupled-MARCM Fas2-deficient clones (*fas2^{eB112}*, *fas2^−* labeled with GFP) and their control Fas2-normal twins (*fas2^+* labeled with mtdTomato) in a wing imaginal disc at puparium formation. The Fas2-deficient clones are missing or consist of single cells, but their *fas2^+* twins can be larger than average WT clones, reminiscent of the Minute effect. (C) *fas2^−* coupled-MARCM clones (labeled with GFP, green) show a grow deficit in imaginal disc derivatives compared to their control twins (mtdTomato, red) but do much better in abdominal histoblasts. Compare the number and size of *fas2^−* clones (GFP) with that of their twin *fas2^+* controls (mtdTomato) in notum vs. abdomen. The picture corresponds to an adult just after eclosion. (D) Coupled-MARCM *fas2^{eB112}* clones (GFP) and twin control clones (mtdTomato) in a wing disc stained with anti-pH3 antibody to reveal mitosis. The red outline marks the rim of 2–3 cells around the Fas2-deficient clone (used to quantify the mitotic index). (E) Notum of an adult individual with the whole left heminotum covered by control clone territory (*sn, fas2^+* outlined in red), but without any detectable Fas2-deficient twin (which would have been labeled with *y f^{36a}, fas2^−*). (F) At least some Fas2-deficient cell clones survived in the adult. The picture shows a *y fas2^{eB112} f^{36a}* clone (green arrow) and its *sn^3* control twin (red arrows point to *sn^3* bristles). The clone was induced in 2^nd instar larva. (G) *y fas2^{eB112} f^{36a} M^+* clones in a *Minute* heterozygous background are able to grow normally and differentiate epidermis in the adult (green arrows point to *y fas2^{eB112} f^{36a}* chaetes). (H) Eye clone deficient for Fas2 induced in a *Minute* genetic background. *fas2^{eB112} M^+* null clones (without GFP) could grow more normally in the $M^−/M^+$ heterozygous background (labeled with *Ubi-GFP*). Genotype: *y fas2^{eB112} FRT18A/Ubi-GFP M(1)O^{sp} FRT18A; hs-FLP/+*. The Minute technique consists in creating normal *Minute^+* clones in a heterozygous *Minute^−/Minute^+* mutant individual. The normal *Minute^+* cells grow faster than the *Minute^−/Minute^+* heterozygous background cells.

Thus, simply reducing the proliferation rate of the cells outside of the *fas2⁻* clones allows them to reach a more normal size. Bar is 50μm. (I) Expression of cleaved Caspase 3 in *fas2⁻* null clones is limited to a fraction of cells (arrow). Bar is 50 μm. (J) Quantification of *fas2⁻* clone size (grey bar) and their control twins (*fas2⁺*, white bar) in the adult notum, and rescue of *fas2⁻* MARCM clones by expression of *UAS-fas2^{GPI}* and *UAS-fas^{TRM}* compared to *fas2⁻M⁺* clones and genetic conditions reducing apoptosis (grey bars). Clones were induced in 1ˢᵗ instar larva. The adult size of *fas2⁻Minute⁺* clones (*fas2⁻M⁺*) in a *Minute⁻/Minute⁺* heterozygous genetic background approached the size of the normal *fas2⁺* control clones. Fas2-deficient MARCM clones (*fas2^{eB112}*, *fas2⁻*) growing in a *Df(3L)H99/+* genetic background displayed a little significant (depending on the statistical test used, *t* or *Mann-Whitney*) normalization of clone size in the adult notum. Expression of the *Drosophila-inhibitor of apoptosis* (*UAS-diap*) in *fas2⁻MARCM* clones (*fas2⁻diap^{GOF}*) had a similar effect, while expression of *UAS-P35* had no effect. Clone size is represented as number of chaetes. N is number of clones. (K) Inhibition of the JNK signaling pathway in *MS1096-GAL4/+; UAS-fas2^{RNAi#34084}/UAS-bsk^{RNAi#32977}* did not cancel the reduction in wing size compared to *MS1096-GAL4/+; UAS-bsk^{RNAi#32977}/+* controls. Wing size in $\mu m^2/10^3$. N is number of individuals.
(TIF)

**S2 Fig. Quantification of the expression of JNK activity reporters.** (A) Left, quantification of TRE-DsRed signal in *en-GAL4 UAS-GFP/TRE-DsRed* (*fas2⁺*) and *en-GAL4 UAS-GFP/TRE-DsRed; UAS-fas2^{RNAi#34084}/+* (*fas2^{RNAi}*) wing imaginal discs. The TRE-DsRed signal was amplified using an anti-RFP antibody, and the signal in the anterior compartment was subtracted to the signal in the Posterior compartment in each disc to normalize for differences in staining. Right, quantification of puc-LacZ signal in *en-GAL4 UAS-GFP/+; puc-LacZ/+* (*fas2⁺*) and *en-GAL4 UAS-GFP/+; UAS-fas2^{RNAi#34084}/puc-LacZ* (*fas2^{RNAi}*) wing imaginal discs. The signal in the anterior compartment was subtracted to the signal in the Posterior compartment in each disc to normalize for differences in staining. N is number of wing imaginal discs. (B) Left, quantification of TRE-DsRed signal in *fas2^{eB112}FRT19A; TRE-DsRed/; Tub-GAL4 UAS-GFP/+* MARCM null cell clones (*fas2⁻*) and *fas2^{eB112} FRT18A; TRE-DsRed/+ Minute⁺* null cell clones (*fas2⁻M⁺*) induced in *fas2^{eB112} FRT18A/Ubi-GFP M(1)O^{sp} FRT18A; TRE-DsRed/+* wing imaginal discs. The signal in the background was subtracted to the signal in the clone in each disc. Right, quantification of puc-LacZ signal in *fas2^{eB112}FRT19A; Tub-GAL4 UAS-GFP/puc-LacZ* MARCM null cell clones (*fas2⁻*) and *fas2^{eB112} FRT18A; puc-LacZ/hs-FLP Minute⁺* null cell clones (*fas2⁻M⁺*) induced in *fas2^{eB112} FRT18A/Ubi-GFP M(1)O^{sp} FRT18A; puc-LacZ/hs-FLP* wing imaginal discs. The signal in the background was subtracted to the signal in the clone in each disc. N is number of clones.
(TIF)

**S3 Fig. Fas2 functions via the EGFR pathway.** (A) Left, WT adult head. Right, the hypomorphic *fas2^{eB112}/fas2^{e76}* combination displayed the absence or size reduction of ocelli, as well as loss of bristles in the dorsal head, an alteration reminiscent of *Egfr torpedo* alleles (*Egfr^{top}*). (B) Quantification of fluorescent ppERK antibody signal (ratio of pixels with a signal level higher than 100, out of 255 levels) in the eye disc of *ey*-driven *fas2^{RNAi}* (#34084) FLP-OUTs (green bar) and their control CyO siblings (white bar). N is number of eye imaginal discs. (C) Left, *ey*-driven *fas2^{RNAi}* (#34084) FLP-OUT. Right, *ey*-driven *fas2^{RNAi}* (#34084) FLP-OUT in an *Egfr^{t1}/+* heterozygous background. (D) Suppressors of the *fas2*-null MARCM-clone phenotype in the adult. MARCM *fas2^{eB112}* clones labeled with *yellow* and *forked* were induced in combinations expressing *UAS*-insertions for components of different growth signaling pathways. Expression of activated components of the EGFR signaling pathway (Ras^{V12}, Raf^{GOF} and PI3K - Dp110-) and over-expression of Yki produced a significant correction in the size of

*fas2⁻* clones in the adult notum. In contrast, expression of activated-FGFR (λHtl), which shares most downstream effectors with EGFR, activated-InR (InR$^{R418P}$), activated-Notch (N$^{IN-TRA}$) and myristoylated-Src did not cause a significant suppression. Clone size is number of marked microchaetes. N is number of clones.
(TIF)

**S4 Fig. JNK is required for the Hippo pathway increased signaling in the *fas2* LOF condition.** Quantification of *ex-LacZ* reporter expression in *en-GAL4/+; fas2$^{RNAi#34084}$* wing imaginal discs. The intensity of expression of the *ex-LacZ* reporter (measured as grey average in the red channel) is similar in the anterior and posterior compartments of each control wing imaginal disc (giving a posterior/anterior signal ratio close to 1.0). Expression of *UAS-fas2$^{RNAi}$* (#34084) in the posterior compartment causes a strong increase of *ex-LacZ* expression compared with the anterior compartment in the same disc (*ex-LacZ* P/A signal ratio). While inhibition of the JNK pathway (*UAS-bsk$^{RNAi}$*, #32977) in the posterior compartment slightly increases *ex-LacZ* P/A signal ratio, it strongly suppresses the increase caused by the inhibition of Fas2 expression. N is number of wing imaginal discs.
(TIF)

**S5 Fig. Fas2 dependence on EGFR function.** (A) Fas2 and EGFR are expressed by all cells in imaginal discs. A Fas2::GFP protein trap [13] shows colocalization with EGFR (ImageJ Colocalization plug-in, Pearson´s correlation: 0,3146). (B) Homozygous *Egfr$^{top1}$* imaginal discs showed a lower expression of Fas2 (red channel, labeled with the anti-Fas2 1D4 antibody) than the imaginal discs from their heterozygous siblings (*Egfr$^{top1}$/CyO, GFP*).
(TIF)

**S6 Fig. Fas2 expression is directly regulated by EGFR.** (A) A Fas2::GFP protein trap line shows expression of Fas2 in all cells of imaginal discs, with a maximum in differentiating retinal cells. (B) FLPOUT *UAS-LacZ* clones (red signal) expressing activated-EGFR (λ*Egfr*) display a dramatic increase in Fas2::GFP expression. Note that the saturation of the Fas2::GFP expression prevents the visualization of the normal Fas2::GFP expression in the other cells of the wing disc (compare to Fig 7A which is stained with the 1D4 antibody that only recognizes the TRM isoforms of Fas2). (C) Wing imaginal disc FLPOUT *UAS-LacZ* clones (red signal) expressing activated-EGFR (λ*Egfr*) plus *bsk$^{RNAi314767}$* display a Fas2::GFP signal similar to activated-EGFR clones. (D) Eye imaginal disc FLPOUT *UAS-LacZ* clones (red signal) expressing activated-EGFR (λ*Egfr*) plus *bsk$^{RNAi#31476}$* display a similarly strong Fas2::GFP signal. Note that the saturation of the Fas2::GFP signal only permits a very faint visualization of the normal Fas2 peak of expression in the differentiating retina.
(TIF)

**S1 Text. List of Strains used for experiments.**
(DOCX)

## Acknowledgments

We thank S. Campuzano and P. Martin for the *w M(1)O$^{sp}$ FRT18A/FM7; hsp70-flp; Dp(1;3) A59/TM6B* and the *puc-LacZ* strains, Christian Klambt for the *fas2::GFP* line and E. Schejter for the *pcDNA3-EGFR* plasmid. We are grateful to S. Baars for technical help in some experiments and D. Ferres-Marco for thoughtful comments on the manuscript. Most primary antibodies were obtained from the Developmental Studies Hybridoma Bank.

## Author Contributions

**Conceptualization:** Luis Garcia-Alonso.

**Data curation:** Luis Garcia-Alonso.

**Formal analysis:** Luis Garcia-Alonso.

**Funding acquisition:** Luis Garcia-Alonso.

**Investigation:** Emma Velasquez, Jose A. Gomez-Sanchez, Emmanuelle Donier, Carmen Grijota-Martinez, Hugo Cabedo, Luis Garcia-Alonso.

**Methodology:** Hugo Cabedo, Luis Garcia-Alonso.

**Project administration:** Luis Garcia-Alonso.

**Resources:** Luis Garcia-Alonso.

**Supervision:** Hugo Cabedo, Luis Garcia-Alonso.

**Validation:** Luis Garcia-Alonso.

**Visualization:** Luis Garcia-Alonso.

**Writing – original draft:** Luis Garcia-Alonso.

**Writing – review & editing:** Luis Garcia-Alonso.

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
