## [Decision Letter · Decision Letter 0]

25 Nov 2021

Dear Dr Garcia-Alonso,

Thank you very much for submitting your Research Article entitled 'Fasciclin 2 engages EGFR in an auto-stimulatory loop to promote imaginal disc cell proliferation in Drosophila' to PLOS Genetics.

The manuscript was fully evaluated at the editorial level and by independent peer reviewers. The reviewers appreciated the attention to an important problem, but raised some substantial concerns about the current manuscript. Based on the reviews, we will not be able to accept this version of the manuscript, but we would be willing to review a much-revised version. We cannot, of course, promise publication at that time.

If you decide to revise the manuscript for further consideration at PLOS Genetics, please aim to resubmit within the next 60 days, unless it will take extra time to address the concerns of the reviewers, in which case we would appreciate an expected resubmission date by email to plosgenetics@plos.org.

[LINK]

We are sorry that we cannot be more positive about your manuscript at this stage. Please do not hesitate to contact us if you have any concerns or questions.

Yours sincerely,

Gregory P. Copenhaver

Editor-in-Chief

PLOS Genetics

Reviewer's Responses to Questions

**Comments to the Authors:**

Reviewer #1: PGENETICS D 21 01398

In this very nice paper, Velasquez et al report a novel function of the Drosophila NCAM homolog Fasciclin2. As NCAM, Drosophila Fasciclin2 is a homophilic cell-cell adhesion protein that as the authors convincingly show is cell autonomously needed to promote cell proliferation in imaginal discs. As validate by a thorough combination of loss and gain of function genetics, and immunohistochemical studies, this proliferative trigger is mediated via the EGF-receptor. Importantly, as the authors demonstrate at the end of their manuscript, the EGF-receptor can physically interact with Fascilin2. Quite interestingly, the authors note that activation of the EGF-receptor leads to an increase in Fasciclin2 expression suggesting the presence of a positive feedback loop sustaining EGF-receptor activation and Fasciclin2 expression.

This paper in particular important as it contrasts previous reports where it was shown that in differentiating Drosophila photoreceptor cells, Fasclin2 and EGF-receptor act in an antagonistic relationship (Mao and Freeman, 2009). As mentioned in the discussion, these two reaction properties may reflect different Fascilin2 concentrations, which will be an interesting question to address in future studies.

The data are documented by beautiful and very clear Figures. Just in Figure5A the authors may consider a revision to show intact, non-tilted brains without any imaginal disc obscuring the imaging.

Very nice and important work showing a new twist in a long studied adhesion protein, that given its strong evolutionary conservation is likely to have some implications for NCAM function, too. I support publication as is.

Reviewer #2: In the paper “Fasciclin 2 engages EGFR in an auto-stimulatory loop to promote imaginal disc cell proliferation in Drosophila” by Velasquez et al., the authors demonstrate a role for Fasciclin 2 in the regulation of cell growth and competition by positive regulation of EGFR signaling in imaginal discs, which is contrary to its previously demonstrated role in retinal differentiation. This paper provides an interesting and novel contribution to the field, with genetic and molecular biology evidence of the interaction between Fas2 and Egfr. This manuscript would be a good fit to PLOS Genetics after a more thorough exploration of pathways that are involved downstream of cell autonomous and cell non-autonomous effects. In addition, although the authors did a remarkable job writing in their non-native language, a few revisions to the text are highly encouraged to improve readability – at times, thanks to the evaluation of so many different pathways, the reader gets slightly confused (for example, see “minor suggestion 2”).

Major suggestions:

1. On Fig 2E, slowing growth via the minute technique appears to partially rescue clone size, but not completely back to wildtype. Are there other effects that are not cell competition that could account for the incomplete rescue?

2. For the apoptosis analysis, the authors should take into consideration the growth size of WT clones made into a “DfH99/+” or “diap1(OE)” background. Since the “rescue” of clone sizes is so minute (and UAS-P35 doesn’t even rescue it), genetic background may be an issue. Also, if the authors truly believe that apoptosis is “a consequence and integral part in the process of cell competition” (page 7), then the conclusion that apoptosis inhibition barely affects clonal size would contradict the authors findings. The authors should address the discrepancy (perhaps by finding evidence of/citing cases of cell competition without any major cell death? Or removing the cell death data?).

3. The JNK analysis is well made and very interesting, but the authors did not quantify the results obtained in Fig 4, which is needed in order to appreciate the strength of the rescue (particularly in Fig 4D). It would also be nice to quantify how often JNK activation is observed under the different reporters and conditions.

4. Do slow-growing clones adjacent to Fas2 deficient cells also have reduced EGFR/Erk signaling, or is the cell-non-autonomous growth deficit mediated by another pathway? When you restore the EGFR signaling or downstream effectors in Fas2- cells, what happens to the neighbor cells?

5. Do Fas2- M+ clones differentiate normally, or can the switch in Fas2 to EGFR-suppression be demonstrated in this tissue?

6. Finally, though very interesting, the authors’ conclusion that Fas2 is also a direct, cell-autonomous target of Egfr could also have another explanation. Is Fas2 a target of Yki? Does JNK inactivation upon Egfr activation in clones rescue the increase in Fas2 seen in Figure 7? These experiments would help elucidate if Fas2 levels in Egfr or yki GOF clones are cell-autonomously promoted by Egfr, or is a result of the JNK/Yki non-autonomous activation.

Minor suggestions:

1. Some of the green/red labels inside panels are hard to read (for example, Fig S1 E-G). The authors could place a black rectangle behind the labels, or move them outside the images.

2. As an example of text that would benefit from some changes in flow/rewriting for clarity, on page 8 the authors start by describing how they will evaluate JNK activity, by mentioning the TRE and puc reporters. Then the authors start describing the results of the TRE results (Fig 3E), go back to explain the puc reporter, then mention results from the previous evaluated theme of apoptosis (caspase 3 stains, Fig 3D), to then finally start describing the puc reporter analysis (Fig 4A). The manuscript would benefit from having this section re-written.

3. The authors should use RNAi identifiers other than BL number in figure legends, as the Bloomington stock center reserves the rights to remove or change numbers.

4. Missing reference to figure 1G

5. It’s not clear whether their Ns are number of clones, number of discs, or number of animals. In cases where clones/discs are counted, number of animals should also be identified.

6. Fig S1 lists N of 2 for “fas2- M+”. If that was indeed the case, the authors should increase significantly the number of clones analyzed.

7. Title of figure 3 should be more descriptive. For example, “loss of Fas2 cell autonomously and non-autonomously activates JNK signaling”

8. The final model would be more useful if it included elements of the pathways tested (EGFR, JNK, etc).

Reviewer #3: Previous studies have revealed interactions between the cell adhesion molecule Fas2 and the EGFR activity during morphogenesis. Thus, fas2 gain of function has been shown to promote EGFR activity during axon growth. In contrast, fas2 loss of function has been shown to cause de-repression of EGFR during retinal differentiation. In addition, experiments in the follicular epithelium of the Drosophila ovary has also revealed a role for Fas2 on regulating cell proliferation. These studies have shown that fas2- follicle cell clones displayed a 2-fold increase in BrdU incorporation and in cells in S-phase compared to controls. In addition, while wild type follicle cells stop dividing by stage 6, elimination of fas2 function resulted in the presence of PH3+ cells in later stages and in a 2-fold increase in the number of Cyclin E positive cells, showing that Fas2 represses follicle cell proliferation. All these results have led to suggest that the cell adhesion molecule Fas2 can regulate epithelial cell proliferation and dynamics acting not only as a localized scaffold but also by communicating signals to the nucleus. In this work, the authors analysed the role of Fas2 in another epithelial context, the Drosophila imaginal discs. In contrast to previous results mentioned above, here, the authors find that Fas2 loss leads to reduced proliferation. In addition, they find that fas2 mutant cells show decreased EGFR activity. In fact, based on genetic interactions and rescue analysis, they propose that Fas2 acts via the EGFR pathway to control cell proliferation. They further show that Fas2 and EGFR co-precipitate in cultured HEK293 cells. They also find that EGFR activity promotes the autonomous expression of Fas2, leading them to propose the existence of a feedback loop between Fas2 expression and EGFR activity in wing imaginal disc cells. Finally, they find that Fas2 loss induces increased JNK and Yki activity.

In general, I think there are some interesting results that challenge previous reported roles of Fas2 in cell proliferation and its interaction with other pathways, such as the EGFR, JNK or Hpo. However, I find the work still at a preliminary stage, there are serios problems with some of the results, as they are presented (see below, major concerns) and more work needs to be done to properly address the role of Fas2 in cell proliferation in the imaginal discs and to explain why Fas2 plays opposite roles in different cellular contexts. In summary, part of the analysis seems to me at a preliminary stage and several of the conclusions reached are not based upon a rigorous analysis. Therefore, I do not consider this work of enough scientific value as to merit publication in PLoS Genetics at its present stage.

Major concerns:

1. The authors use hypomorphic conditions and fas2 RNAis to knockdown fas2 expression in imaginal discs. However, it is not clear to me to what extent these two conditions reduce fas2 levels. Antibody staining with anti-Fas2 antibody would resolve this small issue.

2. To confirm that Fas2 is required for cell proliferation, the authors generate Fas2 deficient clones using the Minute technique. They claim fas2-Minute+ clones displayed normalized size in both the wing disc and the notum. However, while clone size is similar to controls in the notum, this is not the case in the wing disc, where the size of the fas2-Minute+ clones are slightly bigger than fas2- clones alone and much smaller than fas2+ clones. Thus, to me, in contrast to what authors propose, this data suggests that Fas2 might be regulating something else besides cell proliferation in the wing disc cells.

A comment to add here is that, while the data for fas2+ or fas2- clones in the notum is represented as a dot plot, this is not the case for fas2-Minute+ clones. This makes difficult to properly assess the rescue.

3. To study the contribution of apoptosis to the fas2 phenotype, the authors test if suppression of apoptosis rescues the fas2- clones. They found this was the case, although slightly, and in certain conditions. However, this is only tested in the notum and not in the wing disc. Thus, I think the authors should also test the effect of inhibiting apoptosis in the wing disc. In general, I think the author switch between the wing disc and the notum to test different aspects of Fas2 function and I believe they should test all parameters in both contexts, as it is not clear, at least to me, that fas2- clones behave the same in both cellular contexts, since i.e., as mentioned above, fas2-Minute+ clones do not behave the same in the notum, nearly complete size rescue, compare to the wing disc, slight size rescue.

4. Next, the authors analysed the involvement of the JNK pathway in the fas2 loss of function phenotype, by inhibiting its activity in fas2 mutant cells. They found that expression of an RNAi against the JNK basket did not rescue the size of the fas2- clones, suggesting little if any involvement of the JNK pathway in fas2 loss of function.

5. To further test the implication of the JNK pathway, the authors tested the expression of two downstream targets of this pathway in fas2- clones. Surprisingly, there is a big difference in the induction of the 2 downstream targets of the JNK pathway tested, TRE-DsRed and puc-lacZ. Thus, while there is a high increase in the levels of puc-lacZ in fas2 RNAi conditions, TRE-DsRed seems to be only slightly increased. In addition, levels of the two reporters have not been properly quantified. Furthermore, I think authors should provide an explanation for the difference between the expression of the two targets. In addition, the authors claim that a reduction in the dose of puc, as it happens in a puc-lacZ background, leads to widespread apoptosis and strong alterations in the A/P boundary. However, this is only observed in panel B of Fig.4, while the discs shown in panels A and C of this figure, besides having the same genotype, do not show these strong defects. Thus, how often are the defects shown in Fig4B observed? How strong is this interaction? Finally, with respect to this section, I am a bit confused about the contribution of the JNK pathway in the fas2 phenotype, I am not sure whether the authors relate it to apoptosis, although there is no sign of apoptosis in fas2-deficient clones, or growth, which is not clear to me what they mean with growth as they have shown fas2- cells do not change their size. In addition, as bsk RNAi is unable to rescue the fas2 mutant clone size, what is the functional meaning of the elevated expression of the JNK targets in the absence of fas2?

6. Then they test the contribution of EGFR activity to the fas2 loss of function phenotype. In order to do this, they again use the eye disc and the wing disc to test different things. On one hand, they test EGFR activity in fas2- clones in the eye disc, while, on the other hand, they test the ability of activated EGFR pathway to rescue fas2 loss of function in the wing disc. As mentioned above, I think they should all be tested in the same cellular context. In addition, pictures showing EGFR activity are not of enough quality. Thus, even though the internal control for ppERK levels, the ring gland, show similar levels in both experimental and control conditions, which it is in any case difficult to appreciate due to oversaturation of the signal, the general levels of ppERK in the eye disc outside the clones seems much lowered in the experimental condition compared to the control. In addition, there are many GFP positive cells that do not show changes in anti-ppERK staining. Thus, I believe this result, as it is presented, is not conclusive.

7. In the next section, the authors analyse the relationship between Fas2 and the Hippo pathway. In the follicular epithelium it has been shown that Fas2 tumors are suppressed by Wts overexpression, indicating that Fas2 signals through Wts to repress proliferation, although in this case Ex seems to play a minor role. Similarly, here, the authors show an upregulation of Yki activation in fas2- clones. As this is reverted upon JNK repression, the authors propose that indirect JNK activation by loss of fas2 promotes Yki activation. However, they do not find any effect on fas2 loss of function phenotype when suppressing wts. This questions the functional meaning of the elevated expression of ex-lacZ in fas2 clones. I would also be interested to know whether overexpression of wts would rescue loss of fas2 in the wing disc, as it does in the follicular epithelium.

Here, it is also worth mentioning that when analysing the ability of suppressing JNK on the observed activation of the Hpo pathway upon depletion of fas2, the number of samples analysed is only five. I do not think this sample is big enough as to propose a statistically significant difference. This should be better analysed by increasing sample size.

8. Finally, the authors show that Fas2 physically interacts with EGFR in HEK293 cells. This is a heterologous system and this interaction might not be maintained in Drosophila imaginal disc cells. Could the authors test this interaction in Drosophila S2 cells? In addition, if these two proteins interact physically they should co-localize. The authors could test this by immunostaining wing discs from available transgenic flies carrying a superfolder-GFP inserted into the EGFR locus with anti-GFP and anti-Fas2 antibodies.

Minor comments

1. The authors refer to Fig.2B, C when talking about the rescue of fas2- clones with either Fas2GPI or Fas2TM, however these panels only show Fas2GPI.

**Have all data underlying the figures and results presented in the manuscript been provided?**

Reviewer #1: Yes

Reviewer #2: Yes

Reviewer #3: Yes

PLOS authors have the option to publish the peer review history of their article (what does this mean?). If published, this will include your full peer review and any attached files.

Reviewer #1: No

Reviewer #2: No

Reviewer #3: No

---

## [Decision Letter · Decision Letter 1]

28 Apr 2022

Dear Dr Garcia-Alonso,

We are pleased to inform you that your manuscript entitled "Fasciclin 2 engages EGFR in an auto-stimulatory loop to promote imaginal disc cell proliferation in Drosophila" has been editorially accepted for publication in PLOS Genetics. Congratulations!

As you will see below, Reviewer 2 has some very minor textual issues that you can address as you prepare your final draft for the production team, including being explicit that rescue experiment in Figure 2E is partial rather than full.  The editorial team will not need to re-evaluate.

Yours sincerely,

Gregory P. Copenhaver

Editor-in-Chief

PLOS Genetics

Comments from the reviewers (if applicable):

Reviewer's Responses to Questions

**Comments to the Authors:**

Reviewer #2: The authors successfully improved the manuscript, taking into consideration most suggestions by the reviewers. The manuscript is in excellent shape, and we only recommend one careful consideration regarding the "rescue experiment" of Fig2E (see below). All other suggestions are minor and are only recommended for better readership:

---main: the minute rescue experiment (Fig 2E) certainly shows an amelioration of the phenotype, but it was far from being a full rescue. It obviously doesn't need to be a full rescue to be significant, but in the manuscript text the authors should acknowledge there’s an amelioration of the phenotype, not full rescue (or "normalization").

---minor:

-Larval brain referred to as Fig. 2C, which is eye disc according to legends

-Labeling for 4B/C is confusing-- is puc-LacZ/+ genotype or is it supposed to be visible in white stain in the image?

-Labels missing for bottom 2 rows of figure 6 (I think just single color but could use more explanation in the figure)

Reviewer #3: I believe the authors have addressed most of the concerns raised by all reviewers in a manner that makes this work suitable for publication in PLoS genetics. However, and in other to improve explicitness and simplicity, revisisons to the text are encouraged.

**Have all data underlying the figures and results presented in the manuscript been provided?**

Reviewer #2: Yes

Reviewer #3: Yes

PLOS authors have the option to publish the peer review history of their article (what does this mean?). If published, this will include your full peer review and any attached files.

Reviewer #2: No

Reviewer #3: No

**Data Deposition**

http://datadryad.org/submit?journalID=pgenetics&manu=PGENETICS-D-21-01398R1

**Press Queries**

---

## [Editor Report · Acceptance letter]

1 Jun 2022

PGENETICS-D-21-01398R1 

Fasciclin 2 engages EGFR in an auto-stimulatory loop to promote imaginal disc cell proliferation in *Drosophila*

Dear Dr Garcia-Alonso, 

We are pleased to inform you that your manuscript entitled "Fasciclin 2 engages EGFR in an auto-stimulatory loop to promote imaginal disc cell proliferation in *Drosophila*" has been formally accepted for publication in PLOS Genetics! Your manuscript is now with our production department and you will be notified of the publication date in due course.

With kind regards,

Anita Estes

PLOS Genetics

On behalf of:
